# Collaboration! Towards Robust Neural Methods for Routing Problems

**Jianan Zhou**
Nanyang Technological University
`jianan004@e.ntu.edu.sg`

**Yaoxin Wu**[*]
Eindhoven University of Technology
`y.wu2@tue.nl`

**Zhiguang Cao**
Singapore Management University
`zgcao@smu.edu.sg`

**Wen Song**
Shandong University
`wensong@email.sdu.edu.cn`

**Jie Zhang,   Zhiqi Shen**
Nanyang Technological University
`{zhangj,zqshen}@ntu.edu.sg`

## Abstract

Despite enjoying desirable efficiency and reduced reliance on domain expertise, existing neural methods for vehicle routing problems (VRPs) suffer from severe robustness issues – their performance significantly deteriorates on clean instances with crafted perturbations. To enhance robustness, we propose an ensemble-based *Collaborative Neural Framework (CNF)* w.r.t. the defense of neural VRP methods, which is crucial yet underexplored in the literature. Given a neural VRP method, we adversarially train multiple models in a collaborative manner to synergistically promote robustness against attacks, while boosting standard generalization on clean instances. A neural router is designed to adeptly distribute training instances among models, enhancing overall load balancing and collaborative efficacy. Extensive experiments verify the effectiveness and versatility of CNF in defending against various attacks across different neural VRP methods. Notably, our approach also achieves impressive out-of-distribution generalization on benchmark instances.

## 1  Introduction

Combinatorial optimization problems (COPs) are crucial yet challenging to solve due to the NP-hardness. Neural combinatorial optimization (NCO) aims to leverage machine learning (ML) to automatically learn powerful heuristics for solving COPs, and has attracted considerable attention recently [2]. Among them, a large number of NCO works develop neural methods for *vehicle routing problems* (VRPs) – one of the most classic COPs with broad applications in transportation [54], logistics [35], planning and scheduling [52], etc. With various training paradigms (e.g., reinforcement learning (RL)), the neural methods learn construction or improvement heuristics, which achieve competitive or even superior performance to the conventional algorithms. However, recent studies show that these neural methods are plagued by severe robustness issues [20], where their performance drops devastatingly on clean instances (sampled from the training distribution) with crafted perturbations.

Although the robustness issue has been investigated in a couple of recent works [87, 20, 42], the defensive methods on how to help forge sufficiently robust neural VRP methods are still underexplored.

---

[*]Yaoxin Wu is the corresponding author.

38th Conference on Neural Information Processing Systems (NeurIPS 2024).

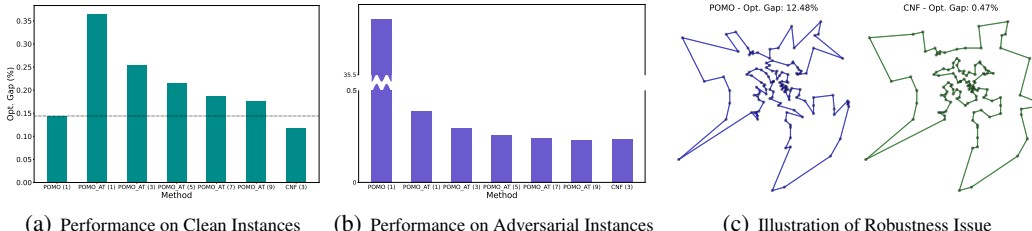

(a) Performance on Clean Instances  (b) Performance on Adversarial Instances  (c) Illustration of Robustness Issue

Figure 1: (a-b) Performance of POMO [38] on TSP100 against the attacker in [87]. The value in brackets denotes the number of trained models. We report the average optimality (opt.) gap over 1000 test instances. (c) Solution visualizations on an adversarial instance. These results reveal the vulnerability of existing neural methods to adversarial attacks, and the existence of undesirable trade-off between standard generalization (a) and adversarial robustness (b) in VRPs. Details of the attacker and experimental setups can be found in Appendix B.1 and Section 5, respectively.

In particular, existing endeavours mainly focus on the *attack*[2] side, where they propose different perturbation models to generate adversarial instances. On the *defense* side, they simply follow the vanilla adversarial training (AT) [48]. Concretely, treated as a *min-max* optimization problem, it first generates adversarial instances that maximally degrade the current model performance, and then minimizes the empirical losses of these adversarial variants. However, vanilla AT is known to face an undesirable trade-off [66, 85] between standard generalization (on clean instances) and adversarial robustness (against adversarial instances). As demonstrated in Fig. 1, vanilla AT improves adversarial robustness of the neural VRP method at the expense of standard generalization (e.g., POMO (1) vs. POMO_AT (1)). One key reason is that the training model is not sufficiently expressive [20]. We empirically justify this viewpoint by increasing the model capacity through ensembling multiple models, which partially alleviates the trade-off. However, it is still an open question on how to effectively synergize multiple models to achieve favorable overall performance on both clean and adversarial instances within a reasonable computational budget.

In this paper, we focus on the *defense* of neural VRP methods, aiming to concurrently enhance both the standard generalization and adversarial robustness. We resort to the ensemble-based AT method to achieve this objective. Instead of separately training multiple models, we propose a *Collaborative Neural Framework (CNF)* to exert AT on multiple models in a collaborative manner. Specifically, in the inner maximization optimization of CNF, we synergize multiple models to further generate the *global* adversarial instance for each clean instance by attacking the best-performing model, rather than only leveraging each model to independently generate their own *local* adversarial instances. In doing so, the generated adversarial instances are diverse and strong in benefiting the policy exploration and attacking the models, respectively (see Section 4). In the outer minimization optimization of CNF, we train an attention-based neural router to forward instances to models for effective training, which helps achieve satisfactory load balancing and collaborative efficacy.

Our contributions are outlined as follows. 1) In contrast to the recent endeavors on the attack side, we concentrate on the defense of neural VRP methods, which is crucial yet underexplored in the literature. We empirically observe that the defense through vanilla AT may lead to the undesirable trade-off between standard generalization and adversarial robustness in VRPs. 2) We propose an ensemble-based collaborative neural framework to concurrently enhance the performance on both clean and adversarial instances. Specifically, we propose to further generate global adversarial instances, and design an attention-based neural router to distribute instances to each model for effective training. 3) We evaluate the effectiveness and versatility of our method against various attacks on different VRPs, such as the symmetric and asymmetric traveling salesman problem (TSP, ATSP) and capacitated vehicle routing problem (CVRP). Results show that our framework can greatly improve the adversarial robustness of neural methods while even boosting the standard generalization. Beyond the expectation, we also observe the improved out-of-distribution (OOD) generalization on both synthetic and benchmark instances, which may suggest the favorable potential of our method in promoting various types of generalization of neural VRP methods.

---

[2]Note that in CO, there may not be an intentional attacker seeking to compromise the model in practice. In this paper, an "attacker" refers to a method of generating instances where the current model underperforms.

## 2 Related Work

**Neural VRP Methods.** Most neural VRP methods learn construction heuristics, which are mainly divided into two categories, i.e., autoregressive and non-autoregressive ones. Autoregressive methods sequentially construct the solution by adding one feasible node at each step. [71] proposes the Pointer Network (Ptr-Net) to solve TSP with supervised learning. Subsequent works train Ptr-Net with RL to solve TSP [1] and CVRP [51]. [37] introduces the attention model (AM) based on the Transformer architecture [69] to solve a wide range of COPs including TSP and CVRP. [38] further proposes the policy optimization with multiple optima (POMO), which improves upon AM by exploiting solution symmetries. Further advancements [39, 34, 3, 24, 13, 44, 27, 45, 21, 12] are often developed on top of AM and POMO. Regarding non-autoregressive methods, the solution is typically constructed in a one-shot manner without iterative forward passing through the model. [31] leverages the graph convolutional network to predict the probability of each edge appearing on the optimal tour (i.e., heat-map) using supervised learning. Recent works [17, 36, 56, 64, 49, 82, 33, 77] further improve its performance and scalability by using advanced models, training paradigms, and search strategies. We refer to [40, 28, 83, 88] for scalability studies, to [30, 5, 90, 18, 41, 16, 89, 4] for generalization studies, and to [11, 59, 84, 61] for other COP studies. On the other hand, some neural methods learn improvement heuristics to refine an initial feasible solution iteratively, until a termination condition is satisfied. In this line of research, the classic local search methods and specialized heuristic solvers for VRPs are usually exploited [8, 43, 10, 76, 47, 78, 46]. In general, the improvement heuristics can achieve better performance than the construction ones, but at the expense of much longer inference time. In this paper, we focused on autoregressive construction methods.

**Robustness of Neural VRP Methods.** There is a recent research trend on the robustness of neural methods for COPs [68, 20, 42], with only a few works on VRPs [87, 20, 42]. In general, they primarily focus on attacking neural construction heuristics by introducing effective perturbation models to generate adversarial instances that are underperformed by the current model. Following the AT paradigm, [87] perturbs node coordinates of TSP instances by solving an inner maximization problem (similar to the fast gradient sign method [22]), and trains the model with a hardness-aware instance-reweighted loss function. [20] proposes an efficient and sound perturbation model, which ensures the optimal solution to the perturbed TSP instance can be directly derived. It adversarially inserts several nodes into the clean instance by maximizing the cross-entropy over the edges, so that the predicted route is maximally different from the derived optimal one. [42] leverages a no-worse optimal cost guarantee (i.e., by lowering the cost of a partial problem) to generate adversarial instances for asymmetric TSP. However, existing methods mainly follow vanilla AT [48] to deploy the defense, leaving a considerable gap to further consolidate robustness.

**Robustness in Other Domains.** Deep neural networks are vulnerable to adversarial examples [22], spurring the development of numerous attack and defensive methods to mitigate the arisen security issue across various domains. 1) **Vision:** Early research on adversarial robustness mainly focus on the continuous image domain (e.g., at the granularity of pixels). The vanilla AT, as formulated by [48] through min-max optimization, has inspired significant advancements in the field [65, 60, 85, 7, 86]. 2) **Language:** This domain investigates how malicious inputs (e.g., characters and words) can deceive (large) language models into making incorrect decisions or producing unintended outcomes [14, 72, 50, 91, 74]. Challenges include the discrete nature of natural languages and the complexity of linguistic structures, necessitating sophisticated techniques for generating and defending against adversarial attacks. 3) **Graph:** Graph neural networks are also susceptible to adversarial perturbations in the underlying graph structures [92], prompting research to enhance their robustness [29, 15, 19, 23, 63]. Similar to the language domain, challenges stem from the discrete nature of graphs and the interconnected nature of graph data. Although various defensive methods have been proposed for these specific domains, most are not adaptable to the VRP (or COP) domain due to their needs for ground-truth labels, reliance on the imperceptible perturbation model, and unique challenges inherent in combinatorial optimization.

## 3 Preliminaries

### 3.1 Neural VRP Methods

**Problem Definition.** Without loss of generality, we define a VRP instance $x$ over a graph $\mathcal{G} = \{\mathcal{V}, \mathcal{E}\}$, where $\mathcal{V} = \{v_i\}_{i=1}^n$ represents the node set, and $(v_i, v_j) \in \mathcal{E}$ represents the edge set with $v_i \neq v_j$.

The solution $\tau$ to a VRP instance is a tour, i.e., a sequence of nodes in $\mathcal{V}$. The cost function $c(\cdot)$ computes the total length of a given tour. The objective is to seek an optimal tour $\tau^*$ with the minimal cost: $\tau^* = \arg\min_{\tau \in \Phi} c(\tau|x)$, where $\Phi$ is the set of all feasible tours which obey the problem-specific constraints. For example, a feasible tour in TSP should visit each node exactly once, and return to the starting node in the end. For CVRP, each customer node in $\mathcal{V}$ is associated with a demand $\delta_i$, and a depot node $v_0$ is additionally added into $\mathcal{V}$ with $\delta_0 = 0$. Given the capacity $Q$ for each vehicle, a tour in CVRP consists of multiple sub-tours, each of which represents a vehicle starting from $v_0$, visiting a subset of nodes in $\mathcal{V}$ and returning to $v_0$. It is feasible if each customer node in $\mathcal{V}$ is visited exactly once, and the total demand in each sub-tour is upper bounded by $Q$. The optimality gap $\frac{c(\tau) - c(\tau^*)}{c(\tau^*)} \times 100\%$ is used to measure how far a solution is from the optimal solution.

**Autoregressive Construction Methods.** Popular neural methods [37, 38] construct a solution to a VRP instance following Markov Decision Process (MDP), where the policy is parameterized by a neural network with parameters $\theta$. The policy takes the states as inputs, which are instantiated by features of the instance and the partially constructed solution. Then, it outputs the probability distribution of valid nodes to be visited next, from which an action is taken by either greedy rollout or sampling. After a complete tour $\tau$ is constructed, the probability of the tour can be factorized via the chain rule as $p_\theta(\tau|x) = \prod_{s=1}^{S} p_\theta(\pi_\theta^s | \pi_\theta^{<s}, x)$, where $\pi_\theta^s$ and $\pi_\theta^{<s}$ represent the selected node and the partial solution at the $s_{\text{th}}$ step, and $S$ is the number of total steps. Typically, the reward is defined as the negative length of a tour $-c(\tau|x)$. The policy network is commonly trained with REINFORCE [75]. With a baseline function $b(\cdot)$ to reduce the gradient variance and stabilize the training, it estimates the gradient of the expected reward as:

$$\nabla_\theta \mathcal{L}(\theta|x) = \mathbb{E}_{p_\theta(\tau|x)}[(c(\tau) - b(x))\nabla_\theta \log p_\theta(\tau|x)]. \tag{1}$$

### 3.2 Adversarial Training

Adversarial training is one of the most effective and practical techniques to equip deep learning models with adversarial robustness against crafted perturbations on the clean instance. In the supervised fashion, where the clean instance $x$ and ground truth (GT) label $y$ are given, AT is commonly formulated as a min-max optimization problem:

$$\min_\theta \mathbb{E}_{(x,y) \sim \mathcal{D}}[\ell(y, f_\theta(\tilde{x}))], \text{ with } \tilde{x} = \arg\max_{\tilde{x}_i \in \mathcal{N}_\epsilon[x]}[\ell(y, f_\theta(\tilde{x}_i))], \tag{2}$$

where $\mathcal{D}$ is the data distribution; $\ell$ is the loss function; $f_\theta(\cdot)$ is the model prediction with parameters $\theta$; $\mathcal{N}_\epsilon[x]$ is the neighborhood around $x$, with its size constrained by the attack budget $\epsilon$. The solution to the inner maximization is typically approximated by projected gradient descent:

$$x^{(t+1)} = \Pi_{\mathcal{N}_\epsilon[x]}[x^{(t)} + \alpha \cdot \texttt{sign}(\nabla_{x^{(t)}} \ell(y, f_\theta(x^{(t)})))], \tag{3}$$

where $\alpha$ is the step size; $\Pi$ is the projection operator that projects the adversarial instance back to the neighborhood $\mathcal{N}_\epsilon[x]$; $x^{(t)}$ is the adversarial instance found at step $t$; and the $\texttt{sign}$ operator is used to take the gradient direction and carefully control the attack budget. Typically, $x^{(0)}$ is initialized by the clean instance or randomly perturbed instance with small Gaussian or Uniform noises. The adversarial instance is updated iteratively towards loss maximization until a stop criterion is satisfied.

**AT for VRPs.** Most ML research on adversarial robustness focuses on the continuous image domain [22, 48]. We would like to highlight two main differences in the context of discrete VRPs (or COPs). 1) *Imperceptible perturbation:* The adversarial instance $\tilde{x}$ is typically generated within a small neighborhood of the clean instance $x$, so that the adversarial perturbation is imperceptible to human eyes. For example, the adversarial instance in image related tasks is typically bounded by $\mathcal{N}_\epsilon[x] : \|x - \tilde{x}\|_p \leq \epsilon$ under the $l_p$ norm threat model. When the attack budget $\epsilon$ is small enough, $\tilde{x}$ retains the GT label of $x$. However, it is not the case for VRPs due to the nature of discreteness. The optimal solution can be significantly changed even if only a small part of the instance is modified. Therefore, the subjective imperceptible perturbation is not a realistic goal in VRPs, and we do not exert such an explicit imperceptible constraint on the perturbation model (see Appendix A for further discussions). In this paper, we set the attack budget within a reasonable range based on the attack methods. 2) *Accuracy-robustness trade-off:* The standard generalization and adversarial robustness seem to be conflicting goals in image related tasks. With increasing adversarial robustness the standard generalization tends to decrease, and a number of works intend to mitigate such a trade-off in the image domain [66, 85, 73, 57, 80]. By contrast, a recent work [20] claims the existence of

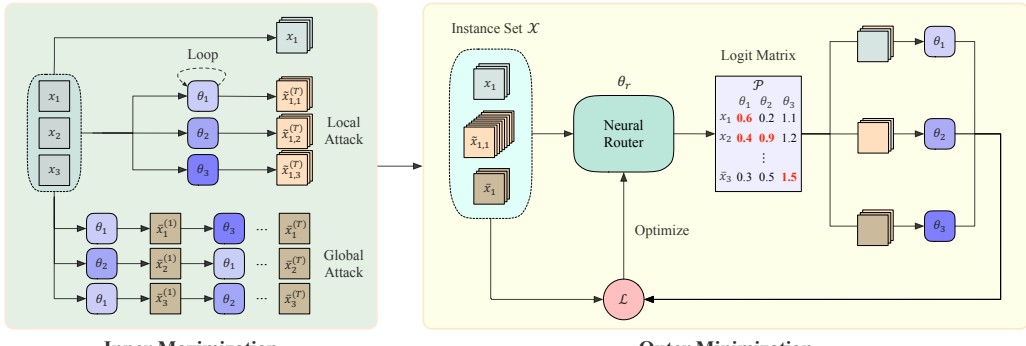

Figure 2: The overview of CNF. Suppose we train $M = 3$ models ($\Theta = \{\theta_1, \theta_2, \theta_3\}$) on a batch ($B = 3$) of clean instances. The inner maximization generates local ($\tilde{x}$) and global ($\bar{x}$) adversarial instances within $T$ steps. In the outer minimization, a neural router $\theta_r$ is jointly trained to distribute instances to the $M$ models for training. Specifically, based on the logit matrix $\mathcal{P}$ predicted by the neural router, each model selects the instances with Top$\mathcal{K}$-largest logits (e.g., red ones). The neural router is optimized to maximize the improvement of collaborative performance after each training step of $\Theta$. For simplicity, we omit the superscripts of instances in the outer minimization.

neural solvers with high accuracy and robustness in COPs. They state that a *sufficiently expressive* model does not suffer from the trade-off given the problem-specific *efficient and sound* perturbation model, which guarantees the correct GT label of the perturbed instance. However, by following the vanilla AT, we empirically observe that the undesirable trade-off may still exist in VRPs (as shown in Fig. 1), which is mainly due to the insufficient model capacity under the specific perturbation model. Furthermore, similar to combinatorial optimization, although the language and graph domains are also discrete, their methods cannot be trivially adapted to VRPs (see Appendix A).

## 4 Collaborative Neural Framework

In this section, we first present the motivation and overview of the proposed framework, and then introduce the technical details. Overall, we propose a collaborative neural framework to synergistically promote adversarial robustness among multiple models, while boosting standard generalization. Since conducting AT for deep learning models from scratch is computationally expensive due to the extra inner maximization steps, we use the model pretrained on clean instances as a warm-start for subsequent AT steps. An overview of the proposed method is illustrated in Fig. 2.

**Motivation.** Motivated by the empirical observations that 1) existing neural VRP methods suffer from severe adversarial robustness issue; 2) undesirable trade-off between adversarial robustness and standard generalization may exist when following the vanilla AT, we propose to adversarially train multiple models in a *collaborative* manner to mitigate the above-mentioned issues within a reasonable computational budget. It then raises the research question on how to effectively and efficiently train multiple models under the AT framework, involving a pair of inner maximization and outer minimization, which will be detailed in the following parts. Note that despite the accuracy-robustness trade-off being a well-known research problem in the literature of adversarial ML, most works focus on the image domain. Due to the needs for GT labels or the dependence on the imperceptible perturbation model, their methods (e.g., TRADES [85], Robust Self-Training with AT [57]) cannot be directly leveraged to solve this trade-off in VRPs. We refer to Appendix A for further discussions.

**Overview.** Given a pretrained model $\theta_p$, CNF deploys the AT on its $M$ copies (i.e., $\Theta^{(0)} = \{\theta_j^{(0)}\}_{j=1}^M \leftarrow \theta_p$) in a collaborative manner. Concretely, at each training step, it first solves the inner maximization optimization to synergistically generate the local ($\tilde{x}$) and global ($\bar{x}$) adversarial instances, on which the current models underperform. Then, in the outer minimization optimization, we jointly train an attention-based neural router $\theta_r$ with all models $\Theta$ by RL in an end-to-end manner. By adaptively distributing instances to different models for training, the neural router can reasonably exploit the overall capacity of models and thus achieve satisfactory load balancing and collaborative efficacy. During inference, we discard the neural router $\theta_r$ and use the trained models $\Theta$ to solve each

**Algorithm 1** Collaborative Neural Framework for VRPs

---

**Input:** training steps: $E$, number of models: $M$, attack steps: $T$, batch size: $B$, pretrained model: $\theta_p$;

**Output:** robust model set $\Theta^{(E)} = \{\theta_j^{(E)}\}_{j=1}^M$;

1: Initialize $\Theta^{(0)} = \{\theta_1^{(0)}, \cdots, \theta_M^{(0)}\} \leftarrow \theta_p, \theta_r^{(0)}$
2: **for** $e = 1, \ldots, E$ **do**
3:     $\{x_i\}_{i=1}^B \leftarrow$ Sample a batch of clean instances
4:     Initialize $\tilde{x}_{i,j}^{(0)}, \bar{x}_i^{(0)} \leftarrow x_i, \ \forall i \in [1, B], \ \forall j \in [1, M]$
5:     **for** t $= 1, \ldots, T$ **do**                           ▷ *Inner Maximization*
6:         $\tilde{x}_{i,j}^{(t)} \leftarrow$ Approximate solutions to $\max \ell(\tilde{x}_{i,j}^{(t-1)}; \theta_j^{(e-1)}), \ \forall i \in [1, B], \ \forall j \in [1, M]$
7:         $\theta_{b_i}^{(e-1)} \leftarrow$ Choose the best-performing model for $\bar{x}_i^{(t-1)}$ from $\Theta^{(e-1)}, \ \forall i \in [1, B]$
8:         $\bar{x}_i^{(t)} \leftarrow$ Approximate solutions to $\max \ell(\bar{x}_i^{(t-1)}; \theta_{b_i}^{(e-1)}), \ \forall i \in [1, B]$
9:     **end for**
10:     $\mathcal{X} \leftarrow \left\{ \{x_i, \tilde{x}_{i,j}^{(T)}, \bar{x}_i^{(T)}\}, \ \forall i \in [1, B], \ \forall j \in [1, M] \right\}$     ▷ *Outer Minimization*
11:     $\mathcal{R} \leftarrow$ Evaluate $\mathcal{X}$ on $\Theta^{(e-1)}$
12:     $\tilde{\mathcal{P}} \leftarrow \text{Softmax}(f_{\theta_r^{(e-1)}}(\mathcal{X}, \mathcal{R}))$
13:     $\Theta^{(e)} \leftarrow$ Train $\theta_j^{(e-1)} \in \Theta^{(e-1)}$ on $\text{Top}\mathcal{K}(\tilde{\mathcal{P}}_{\cdot j})$ instances, $\ \forall j \in [1, M]$
14:     $\mathcal{R}' \leftarrow$ Evaluate $\mathcal{X}$ on $\Theta^{(e)}$
15:     $\theta_r^{(e)} \leftarrow$ Update neural router $\theta_r^{(e-1)}$ with the gradient $\nabla_{\theta_r^{(e-1)}} \mathcal{L}$ using Eq. (6)
16: **end for**

---

instance. The best solution among them is returned to reflect the final *collaborative performance*. We present the pseudocode of CNF in Algorithm 1, and elaborate each part in the following subsections.

## 4.1 Inner Maximization

The inner maximization aims to generate adversarial instances for the training in the outer minimization, which should be 1) effective in attacking the framework; 2) diverse to benefit the policy exploration for VRPs. Typically, an iterative attack method generates local adversarial instances for each model only based on its own parameter (e.g., the same $\theta$ in Eq. (3) is repetitively used throughout the generation). Such *local attack* (line 6) only focuses on degrading each individual model, failing to consider the ensemble effect of multiple models. Due to the existence of multiple models in CNF, we are motivated to further develop a general form of local attack – *global attack* (line 7-8), where each adversarial instance can be generated using different model parameters within $T$ generation steps. Concretely, given each clean instance $x$, we generate the global adversarial instance $\bar{x}$ by attacking the corresponding *best-performing model* in each iteration of the inner maximization. In doing so, compared with the sole local attack, the generated adversarial instances are more diverse to successfully attack the models $\Theta$ (see Appendix A for further discussions). Without loss of generality, we take the attacker from [87] as an example, which directly maximizes the variant of the reinforcement loss as follows:

$$\ell(x; \theta) = \frac{c(\tau)}{b(x)} \log p_\theta(\tau|x), \tag{4}$$

where $b(\cdot)$ is the baseline function (as shown in Eq. (1)). On top of it, we generate the global adversarial instance $\bar{x}$ such that:

$$\bar{x}^{(t+1)} = \Pi_{\mathcal{N}}[\bar{x}^{(t)} + \alpha \cdot \nabla_{\bar{x}^{(t)}} \ell(\bar{x}^{(t)}; \theta_b^{(t)})], \quad \theta_b^{(t)} = \arg\min_{\theta \in \Theta} c(\tau|\bar{x}^{(t)}; \theta), \tag{5}$$

where $\bar{x}^{(t)}$ is the global adversarial instance and $\theta_b^{(t)}$ is the best-performing model (w.r.t. $\bar{x}^{(t)}$) at step $t$. If $\bar{x}^{(t)}$ is out of the range, it would be projected back to the valid domain $\mathcal{N}$ by $\Pi$, such as the min-max normalization for continuous variables (e.g., node coordinates) or the rounding operation for discrete variables (e.g., node demands). We discard the sign operator in Eq. (3) to relax the imperceptible constraint. More details are presented in Appendix B.1.

In summary, the local attack is a special case of the global attack, where the same model is chosen as $\theta_b$ in each iteration. While the local attack aims to degrade a single model $\theta$, the global attack can be viewed as explicitly attacking the collaborative performance of all models $\Theta$, which takes into consideration the ensemble effect by attacking $\theta_b$. In CNF, we involve adversarial instances that

are generated by both the local and global attacks to pursue better adversarial robustness, while also including clean instances to preserve standard generalization.

## 4.2 Outer Minimization

After the adversarial instances are generated by the inner maximization, we collect a set of instances $\mathcal{X}$ with $|\mathcal{X}| = N$, which includes clean instances $x$, local adversarial instances $\tilde{x}$ and global adversarial instances $\bar{x}$, to train $M$ models. Here a key problem is that *how are the instances distributed to models for their training, so as to achieve satisfactory load balancing (training efficiency) and collaborative performance (effectiveness)?* To solve this, we design an attention-based neural router, and jointly train it with all models $\Theta$ to maximize the improvement of collaborative performance.

Concretely, we first evaluate each model on $\mathcal{X}$ to obtain a cost matrix $\mathcal{R} \in \mathbb{R}^{N \times M}$. The attention-based neural router $\theta_r$ takes as inputs the instances $\mathcal{X}$ and $\mathcal{R}$, and outputs a logit matrix $f_{\theta_r}(\mathcal{X}, \mathcal{R}) = \mathcal{P} \in \mathbb{R}^{N \times M}$, where $f$ is the decision function. Then, we apply `Softmax` function along the first dimension of $\mathcal{P}$ to obtain the probability matrix $\tilde{\mathcal{P}}$, where the entity $\tilde{\mathcal{P}}_{ij}$ represents the probability of the $i_{\text{th}}$ instance being selected for the outer minimization of the $j_{\text{th}}$ model. For each model, the neural router distributes the instances with Top$\mathcal{K}$-largest predicted probabilities as a batch for its training (line 10-13). In doing so, all models have the same amount ($\mathcal{K}$) of training instances, which explicitly ensures the *load balancing* (see Appendix A). We also discuss other strategies of instance distributing, such as sampling, instance-based choice, etc. More details can be found in Section 5.2.

After all models $\Theta$ are trained with the distributed instances, we further evaluate each model on $\mathcal{X}$, obtaining a new cost matrix $\mathcal{R}' \in \mathbb{R}^{N \times M}$. To pursue desirable *collaborative performance*, it is expected that the neural router $\theta_r$ can reasonably exploit the overall capacity of models. Since the action space of $\theta_r$ is huge and the models $\Theta$ are changing throughout the training, we resort to the reinforcement learning (based on trial-and-error) to optimize parameters of the neural router $\theta_r$ (line 14-15). Specifically, we set $(\min \mathcal{R} - \min \mathcal{R}')$ as the reward signal, and update $\theta_r$ by gradient ascent to maximize the expected return with the following approximation:

$$\nabla_{\theta_r} \mathcal{L}(\theta_r | \mathcal{X}) = \mathbb{E}_{j \in [1,M], i \in \text{Top}\mathcal{K}(\tilde{\mathcal{P}}_{\cdot j}), \tilde{\mathcal{P}}} [(\min \mathcal{R} - \min \mathcal{R}')_i \nabla_{\theta_r} \log \tilde{\mathcal{P}}_{ij}], \tag{6}$$

where the $\min$ operator is applied along the last dimension of $\mathcal{R}$ and $\mathcal{R}'$, since we would like to maximize the improvement of collaborative performance after training with the selected instances. Intuitively, if an entity in $(\min \mathcal{R} - \min \mathcal{R}')$ is positive, it means that, after training with the selected instances, the collaborative performance of all models on the corresponding instance is increased. Thus, the corresponding action taken by the neural router should be reinforced, and vice versa. For example, in Fig. 2, if the reward entity for the first instance $x_1$ is positive, the probability of this action (i.e., the red one in the first row of $\mathcal{P}$) will be reinforced after optimization. Note that the unselected (e.g., black ones) will be masked out in Eq. (6). In doing so, the neural router learns to effectively distribute instances and reasonably exploit the overall model capacity, such that the collaborative performance of all models can be enhanced after optimization. Further details on the neural router structure and the in-depth analysis of the learned routing policy are presented in Appendix C.

## 5 Experiments

In this section, we empirically verify the effectiveness and versatility of CNF against attacks specialized for VRPs, and conduct further analyses to provide the underlying insights. Specifically, our experiments focus on two attack methods [87, 42], since the accuracy-robustness trade-off exists when conducting vanilla AT to defend against them. We conduct the main experiments on POMO [38] with the attacker in [87], and further demonstrate the versatility of the proposed framework on MatNet [39] with the attacker in [42]. More details on the experimental setups, data generation and additional empirical results (e.g., evaluation on large-scale instances) are presented in Appendix D. All experiments are conducted on a machine with NVIDIA V100S-PCIE cards and Intel Xeon Gold 6226 CPU at 2.70GHz. The source code is available at `https://github.com/RoyalSkye/Routing-CNF`.

**Baselines.** 1) *Traditional methods:* we solve TSP instances by Concorde and LKH3 [25], and CVRP instances by hybrid genetic search (HGS) [70] and LKH3. 2) *Neural methods:* we compare our method with the pretrained base model POMO ($\sim$1M parameters), and its variants training with various defensive methods, such as the vanilla adversarial training (POMO_AT), the defensive method

Table 1: Performance evaluation over 1K test instances. The bracket includes the number of models.

| | | n = 100 | | | | | | n = 200 | | | | | |
|---|---|---|---|---|---|---|---|---|---|---|---|---|---|
| | | Uniform (100) | | Fixed Adv. (100) | | Adv. (100) | | Uniform (200) | | Fixed Adv. (200) | | Adv. (200) | |
| | | Gap | Time | Gap | Time | Gap | Time | Gap | Time | Gap | Time | Gap | Time |
| **TSP** | Concorde | 0.000% | 0.3m | 0.000% | 0.3m | – | – | 0.000% | 0.6m | 0.000% | 0.6m | – | – |
| | LKH3 | 0.000% | 1.3m | 0.002% | 2.1m | – | – | 0.000% | 3.9m | 0.005% | 5.8m | – | – |
| | POMO (1) | 0.144% | 0.1m | 35.803% | 0.1m | 35.803% | 0.1m | 0.736% | 0.5m | 63.477% | 0.5m | 63.477% | 0.5m |
| | POMO_AT (1) | 0.365% | 0.1m | 0.390% | 0.1m | 0.330% | 0.1m | 2.151% | 0.5m | 1.248% | 0.5m | 1.154% | 0.5m |
| | POMO_AT (3) | 0.255% | 0.3m | 0.295% | 0.3m | 0.243% | 0.3m | 1.884% | 1.5m | 1.090% | 1.5m | 1.011% | 1.5m |
| | POMO_HAC (3) | 0.135% | 0.3m | 0.344% | 0.3m | 0.316% | 0.3m | 0.683% | 1.5m | 1.308% | 1.5m | 1.273% | 1.5m |
| | POMO_DivTrain (3) | 0.255% | 0.3m | 0.297% | 0.3m | 0.254% | 0.3m | 1.875% | 1.5m | 1.093% | 1.5m | 1.026% | 1.5m |
| | CNF_Greedy (3) | 0.187% | 0.3m | 0.314% | 0.3m | 0.280% | 0.3m | 0.868% | 1.5m | 1.108% | 1.5m | 1.096% | 1.5m |
| | CNF (3) | **0.118%** | 0.3m | **0.236%** | 0.3m | **0.217%** | 0.3m | **0.614%** | 1.5m | **0.954%** | 1.5m | **0.952%** | 1.5m |
| **CVRP** | HGS | 0.000% | 6.6m | 0.000% | 14.6m | – | – | 0.000% | 0.4h | 0.000% | 1.2h | – | – |
| | LKH3 | 0.538% | 18.1m | 0.344% | 23.0m | – | – | 1.116% | 0.5h | 0.761% | 0.6h | – | – |
| | POMO (1) | 1.209% | 0.1m | 3.983% | 0.1m | 3.983% | 0.1m | 2.122% | 0.6m | 16.173% | 0.8m | 16.173% | 0.8m |
| | POMO_AT (1) | 1.456% | 0.1m | 0.882% | 0.1m | 0.935% | 0.1m | 3.249% | 0.6m | 1.384% | 0.6m | 1.435% | 0.6m |
| | POMO_AT (3) | 1.256% | 0.3m | 0.767% | 0.3m | 0.809% | 0.3m | 2.919% | 1.8m | 1.253% | 1.8m | 1.296% | 1.8m |
| | POMO_HAC (3) | 1.085% | 0.3m | 0.829% | 0.3m | 0.848% | 0.3m | 1.974% | 1.8m | 1.374% | 1.8m | 1.367% | 1.8m |
| | POMO_DivTrain (3) | 1.254% | 0.3m | 0.754% | 0.3m | 0.809% | 0.3m | 2.946% | 1.8m | 1.220% | 1.8m | 1.302% | 1.8m |
| | CNF_Greedy (3) | 1.112% | 0.3m | 0.785% | 0.3m | 0.821% | 0.3m | **1.969%** | 1.8m | 1.316% | 1.8m | 1.353% | 1.8m |
| | CNF (3) | **1.073%** | 0.3m | **0.730%** | 0.3m | **0.769%** | 0.3m | 2.031% | 1.8m | **1.193%** | 1.8m | **1.198%** | 1.8m |

⁻ For traditional methods, *Adv.* is not shown since the test adversarial dataset is different for each neural method.

proposed by the attacker [87] (POMO_HAC), and the diversity training [32] from the literature of ensemble-based adversarial ML (POMO_DivTrain). Specifically, POMO_AT adversarially trains the models by first generating local adversarial instances in the inner maximization, and then minimizing their empirical risks in the outer minimization. POMO_HAC further improves the outer minimization by optimizing a hardness-aware instance-reweighted loss function on a mixed dataset, including both clean and local adversarial instances. POMO_DivTrain improves the ensemble diversity by minimizing the cosine similarity between the gradients of models w.r.t. the input. Furthermore, we also compare our method with CNF_Greedy by replacing the neural router with a heuristic greedy selection method, and another advanced ensemble-based AT method TRS [81] (see Appendix D.6). More implementation details of baselines are provided in Appendix D.1.

**Training Setups.** CNF starts with a pretrained model, and then adversarially trains its $M$ copies in a collaborative way. We consider two scales of training instances $n \in \{100, 200\}$. For the pretraining stage, the model is trained on clean instances following the uniform distribution. We use the open-source pretrained POMO for $n = 100$, and retrain the model for $n = 200$. Following the original training setups from [38], Adam optimizer is used with the learning rate of $1e - 4$, the weight decay of $1e - 6$ and the batch size of $B = 64$. To achieve full convergence, we pretrain the model on 300M and 100M clean instances for TSP200 and CVRP200, respectively. After obtaining the pretrained model, we use it to initialize $M = 3$ models, and further adversarially train them on 5M and 2.5M instances for $n = 100$ and $n = 200$, respectively. To save the GPU memory, we reduce the batch size to $B = 32$ for $n = 200$. The optimizer setting is the same as the one employed in the pretraining stage, except that the learning rate is decayed by 10 for the last 40% training instances. For the mixed data collection, we collect $B$ clean instances, $MB$ local adversarial instances and $B$ global adversarial instances in each training step.

**Inference Setups.** For neural methods, we use the greedy rollout with x8 instance augmentations following [38]. We report the average gap over the test dataset containing 1K instances. Concretely, the gap is computed w.r.t. the traditional VRP solvers (i.e., Concorde for TSP, and HGS for CVRP). If multiple models exist, we report the collaborative performance, where the best gap among all models is recorded for each instance. The reported time is the total time to solve the entire dataset. We consider three evaluation metrics: 1) *Uniform (standard generalization):* the performance on clean instances whose distributions are the same as the pretraining ones; 2) *Fixed Adv. (adversarial robustness):* the performance on adversarial instances generated by attacking the pretrained model. It mimics the black-box setting, where the attacker generates adversarial instances using a surrogate model due to the inaccessibility to the current model and the transferability of adversarial instances; 3) *Adv. (adversarial robustness):* the performance on adversarial instances generated by attacking the current model. For a neural method with multiple ($M$) models, it generates $M$K adversarial instances, from which we randomly sample 1K instances to construct the test dataset. It is the conventional white-box metric used to evaluate adversarial robustness in the literature of AT.

## 5.1 Performance Evaluation

The results are shown in Table 1. We have conducted t-test with the threshold of 5% to verify the statistical significance. For all neural methods, we report the inference time on a single GPU. More-

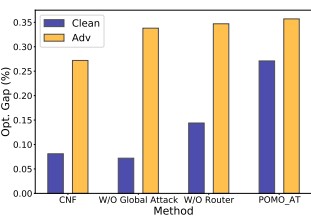 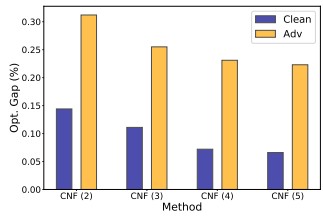 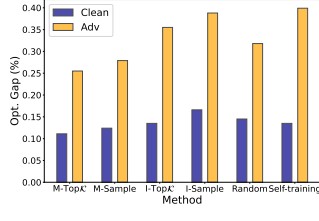

(a) Ablation study on Components  (b) Ablation study on Hyperparameters  (c) Ablation study on Routing Strategies

Figure 3: Ablation studies on TSP100. The metrics of *Uniform* and *Fixed Adv.* are reported.

over, for neural methods with multiple ($M$) models (e.g., CNF (3)), we develop an implementation of parallel evaluation on multiple GPUs, which can further reduce their inference time by almost $M$ times. From the results, we observe that 1) traditional VRP methods are relatively more robust than neural methods against crafted perturbations, demonstrating the importance and necessity of improving adversarial robustness for neural methods; 2) the evaluation metrics of Fixed Adv. and Adv. are almost consistent in the context of VRPs; 3) our method consistently outperforms baselines, and achieves high standard generalization and adversarial robustness concurrently. For CNF, we show the performance of each model on TSP100 in Fig. 5. Although not all models excel well on both clean and adversarial instances, the collaborative performance is quite good, demonstrating diverse expertise of models and the capability of CNF in reasonably exploiting the overall model capacity.

## 5.2 Ablation Study

We conduct extensive ablation studies on TSP100 to demonstrate the effectiveness and sensitivity of our method. Note that the setups are slightly different from the training ones (e.g., half training instances). The detailed results and setups are presented in Fig. 3 and Appendix D.1, respectively.

**Ablation on Components.** We investigate the role of each component in CNF by removing them separately. As demonstrated in Fig. 3(a), despite both components contribute to the collaborative performance, the neural router exhibits a bigger effect due to its favorable potentials to elegantly exploit training instances and model capacity, especially in the presence of multiple models.

**Ablation on Hyperparameters.** We investigate the effect of the number of trained models, which is a key hyperparameter of our method, on the collaborative performance. The results are shown in Fig. 3(b), where we observe that increasing the number of models can further improve the collaborative performance. However, we use $M = 3$ in the main experiments due to the trade-off between empirical performance and computational complexity. We refer to Appendix D.4 for more results.

**Ablation on Routing Strategies.** We further discuss different routing strategies, including neural and heuristic ones. Specifically, given the logit matrix $\mathcal{P}$ predicted by the neural router, there are various ways to distribute instances: 1) *Model choice with TopK (M-TopK):* each model chooses potential instances with TopK-largest logits, which is the default strategy ($\mathcal{K} = B$) in CNF; 2) *Model choice with sampling (M-Sample):* each model chooses potential instances by sampling from the probability distribution (i.e., scaled logits); 3-4) *Instance choice with TopK/sampling (I-TopK/I-Sample):* in contrast to the model choice, each instance chooses potential model(s) either by TopK or sampling. The probability matrix $\tilde{\mathcal{P}}$ is obtained by taking Softmax along the first and last dimension of $\mathcal{P}$ for model choice and instance choice, respectively. Unlike the model choice, instance choice cannot guarantee load balancing. For example, the majority of instances may choose a dominant model (if exists), leaving the remaining models underfitting and therefore weakening the ensemble effect and collaborative performance; 5) *Random:* instances are randomly distributed to each model; 6) *Self-training:* each model is trained on adversarial instances generated by itself without instance distributing. The results in Fig. 3(c) show that M-TopK performs the best.

## 5.3 Out-of-Distribution Generalization

In contrast to other domains (e.g., vision), the set of valid problems is not just a low-dimensional manifold in a high-dimensional space, and hence the manifold hypothesis [62] does not apply to VRPs (or COPs). Therefore, it is critical for neural methods to perform well on adversarial instances when striving for a broader out-of-distriburion (OOD) generalization in VRPs. In this section, we

Table 2: Generalization evaluation on synthetic TSP datasets. Models are only trained on $n = 100$.

| | Cross-Distribution | | | | Cross-Size | | | | Cross-Size & Distribution | | | |
| | Rotation (100) | | Explosion (100) | | Uniform (50) | | Uniform (200) | | Rotation (200) | | Explosion (200) | |
| | Gap | Time | Gap | Time | Gap | Time | Gap | Time | Gap | Time | Gap | Time |
|---|---|---|---|---|---|---|---|---|---|---|---|---|
| Concorde | 0.000% | 0.3m | 0.000% | 0.3m | 0.000% | 0.2m | 0.000% | 0.6m | 0.000% | 0.6m | 0.000% | 0.6m |
| LKH3 | 0.000% | 1.2m | 0.000% | 1.2m | 0.000% | 0.4m | 0.000% | 3.9m | 0.000% | 3.3m | 0.000% | 3.5m |
| POMO (1) | 0.471% | 0.1m | 0.238% | 0.1m | 0.064% | 0.1m | 1.658% | 0.5m | 2.936% | 0.5m | 2.587% | 0.5m |
| POMO_AT (1) | 0.640% | 0.1m | 0.364% | 0.1m | 0.151% | 0.1m | 2.667% | 0.5m | 3.462% | 0.5m | 2.989% | 0.5m |
| POMO_AT (3) | 0.508% | 0.3m | 0.263% | 0.3m | 0.085% | 0.1m | 2.362% | 1.5m | 3.176% | 1.5m | 2.688% | 1.5m |
| POMO_HAC (3) | 0.204% | 0.3m | 0.107% | 0.3m | 0.038% | 0.1m | 1.414% | 1.5m | 2.184% | 1.5m | 1.718% | 1.5m |
| POMO_DivTrain (3) | 0.502% | 0.3m | 0.255% | 0.3m | 0.078% | 0.1m | 2.356% | 1.5m | 3.176% | 1.5m | 2.707% | 1.5m |
| CNF (3) | **0.193%** | 0.3m | **0.084%** | 0.3m | **0.036%** | 0.1m | **1.383%** | 1.5m | **2.055%** | 1.5m | **1.672%** | 1.5m |

further evaluate the OOD generalization performance on unseen instances from both synthetic and benchmark datasets. The empirical results demonstrate that raising robustness against attacks through CNF can favorably promotes various forms of generalization, indicating the potential existence of neural VRP solvers with high generalization and robustness concurrently. The data generation and comprehensive results can be found in Appendix D.

**Synthetic Datasets.** We consider three generalization settings, i.e., cross-distribution, cross-size, and cross-size & distribution. The results are shown in Table 2, from which we observe that simply conducting the vanilla AT somewhat hurts the OOD generalization, while CNF can significantly improve it. Since adversarial robustness is known as a kind of local generalization property [22, 48], the improvements in OOD generalization can be viewed as a byproduct of defending against adversarial attacks and balancing the accuracy-robustness trade-off.

**Benchmark Datasets.** We further evaluate all neural methods on the real-world benchmark datasets, such as TSPLIB [58] and CVRPLIB [67]. We choose representative instances within the range of $n \in [100, 1002]$. The results, presented in Tables 9 and 10, demonstrate that our method performs well across most instances. Note that all neural methods are only trained on $n = 100$.

## 5.4 Versatility Study

To demonstrate the versatility of CNF, we extend its application to MatNet [39] to defend against another attacker in [42]. Specifically, it constructs the adversarial instance by lowering the cost of a partial clean asymmetric TSP (ATSP) instance. When adhering to the vanilla AT, the undesirable trade-off is also observed in the empirical results of [42]. In contrast, our method enables models to achieve both high standard generalization and adversarial robustness. The detailed attack method, training setups, and empirical results are presented in Appendix B.3, D.1 and D.3, respectively.

## 6 Conclusion

This paper studies the crucial yet underexplored adversarial defense of neural VRP methods, filling the gap in the current literature on this topic. We propose an ensemble-based collaborative neural framework to concurrently enhance performance on clean and adversarial instances. Extensive experiments demonstrate the effectiveness and versatility of our method, highlighting its potential to defend against attacks while promoting various forms of generalization of neural VRP methods. Additionally, our work can be viewed as advancing the generalization of neural methods through the lens of adversarial robustness, shedding light on the possibility of building more robust and generalizable neural VRP methods in practice.

The limitation of this work is the increased computational complexity due to the need to synergistically train multiple models. Fortunately, based on experimental results, CNF with just three models has already achieved commendable performance. It even surpasses vanilla AT trained with nine models, demonstrating a better trade-off between empirical performance and computational complexity.

Interesting future research directions may include: 1) designing efficient and effective attack or defense methods for other COPs; 2) pursuing better robustness with fewer computation, such as through conditional computation and parameter sharing; 3) theoretically analyzing the robustness of neural VRP methods, such as certified robustness; 4) investigating the potential of large language models to robustly approximate optimal solutions to COP instances.

## Acknowledgments and Disclosure of Funding

This research is supported by the National Research Foundation, Singapore under its AI Singapore Programme (AISG Award No: AISG3-RP-2022-031). It is also partially supported by Alibaba Group through Alibaba-NTU Singapore Joint Research Institute (JRI), Nanyang Technological University, Singapore. We would like to thank the anonymous reviewers and (S)ACs of NeurIPS 2024 for their constructive comments and dedicated service to the community.

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

# Collaboration! Towards Robust Neural Methods
# for Routing Problems (Appendix)

## A  Frequently Asked Questions

**Load Balancing.** In this paper, load balancing refers to distributing each model with a similar or the same number of training instances from the instance set $\mathcal{X}$, in each outer minimization step. It can avoid the appearance of dominant model or biased performance. The proposed neural router with the Top$\mathcal{K}$ operator explicitly ensures such load balancing since each model is assigned exactly $\mathcal{K}$ instances based on the probability matrix predicted by the neural router.

**Why does CNF Work?** Instead of simply training multiple models, the effectiveness of the proposed collaboration mechanism in CNF can be attributed to its diverse adversarial data generation and the reasonable exploit of overall model capacity. As shown in the ablation study (Fig. 3(a)), the diverse adversarial data generation is helpful in further improving the adversarial robustness (see results of CNF vs. W/O Global Attack). Meanwhile, the neural router has a bigger effect in mitigating the trade-off (see results of CNF vs. W/O Router). Intuitively, by distributing instances to suitable submodels for training, each submodel might be stimulated to have its own expertise. Accordingly, the overlap of their vulnerability areas may be decreased, which could promote the collaborative performance of CNF. As shown in Fig. 5, not all models perform well on each kind of instance. The expertise of $\theta_0$ lies primarily in handling clean instances, whereas $\theta_1$ specializes in dealing with adversarial instances. Such diversity in submodels contributes to the collaborative efficacy of $\Theta$ and the mitigation of the undesirable trade-off between standard generalization and adversarial robustness, thereby significantly outperforming vanilla AT with multiple models.

**Why using Best-performing Model for Global Attack?** The collaborative performance of our framework depends on the best-performing model $\theta_b$ w.r.t. each instance, since its solution will be chosen as the final solution during inference. The goal of inner maximization is to construct the adversarial instance that can successfully fool the framework. Intuitively, if we choose to attack other models (rather than $\theta_b$), the constructed adversarial instances may not successfully fool the best-performing model $\theta_b$, and therefore the final solution to the adversarial instance could still be good, which contradicts the goal of inner maximization. Therefore, to increase the success rate of attacking the framework and generate more diverse adversarial instances, for each instance, we choose the corresponding best-performing model $\theta_b$ as the global model in each attack iteration.

**Divergence of Same Initial Models.** We take POMO [38] as an example. During training, in each step of solution construction, the decoder of the neural solver selects the valid node to be visited by sampling from the probability distribution, rather than using the argmax operation. Even though we initialize all models using the same pretrained model, given the same attack hyperparameters (e.g., attack iterations), the adversarial instances generated by different models are generally not the same at the beginning of the training. Therefore, the same initial models can diverge to different models due to the training on different (adversarial) instances.

**Larger Model.** In addition to training multiple models, increasing the number of parameters for a single model is another way of enhancing the overall model capacity. However, technically, 1) a larger model needs more GPU memory, which puts more requirements on a single GPU device. It is also more sophisticated to enable parallel training on multiple GPUs compared to the counterpart with multiple models; 2) currently, our method conducts AT upon the pretrained model, but there does not exist a larger pretrained model (e.g., larger POMO) in the literature. Despite the technical issues, we try to pretrain a larger POMO (i.e., 18 encoder layers with 3.64M parameters in total) on the uniform distributed data, and further conduct the vanilla AT. The performance is around 0.335% and 0.406% on clean and adversarial instances, respectively, which is inferior to the counterpart with multiple models (i.e., POMO_AT (3)). The superiority of multiple models may be attributed to its ensemble effect and the capacity in learning multiple diverse policies.

**Advanced AT Methods in Other Domains.** In this paper, we mainly focus on the vanilla AT [48]. More advanced AT techniques, such as TRADES [85], AT in RL [55], and ensemble-based AT [65, 32, 53, 79, 81] can be further considered. However, some of them may not be applicable to the VRP domain due to their needs for ground-truth labels or the dependence on the imperceptible perturbation model. 1) TRADES is empirically effective for trading adversarial robustness off against standard generalization on the image classification task. Its loss function is formulated as $\mathcal{L} = \text{CE}(f(x), y) + \beta \text{KL}(f(x), f(\tilde{x}))$, where CE is cross-entropy; KL is KL-divergence; $x$ is a clean instance; $\tilde{x}$ is an adversarial instance; $f(x)$ is the logit predicted by the model; $y$ is the ground-truth label; $\beta$ is a hyperparameter. By explicitly making the outputs of the network (logits) similar for $x$ and $\tilde{x}$, it can mitigate the accuracy-robustness trade-off. However, the above statement is contingent upon the imperceptible perturbation model, where the ground-truth labels of $x$ and $\tilde{x}$ are kept the same. As we discussed in Section 3.2, in the discrete VRPs, the perturbation model does not have such an imperceptible constraint, and the optimal solutions to $x$ and $\tilde{x}$ are not the same in the general case. Therefore, it does not make sense to make the outputs of the model similar for $x$ and $\tilde{x}$. 2) Another interesting direction is AT in RL, where the focus is the attack side rather than the defense side (e.g., most of the design in [55] focuses on the adversarial agent). Specifically, it jointly trains another agent (i.e., the attacker or adversary), whose objective is to impede the first agent, to generate hard trajectories in a two-player zero-sum way. Its goal is to learn a policy that is robust to modeling errors in simulation or mismatch between training and test scenarios. In contrast, our work focuses on the defense side and aims to mitigate the trade-off between standard generalization and adversarial robustness. Moreover, this method is specific to RL while our framework has the potential to work with the supervised learning setting. Overall, it is non-trivial to directly apply this method to address our research problem (e.g., the trade-off). But it is an interesting future research direction to design attack methods specific to RL (e.g., by training another adversarial agent or attacking each step of MDP). 3) Similar to the proposed CNF, ensemble-based AT also leverages multiple models, but with a different motivation (e.g., reducing the adversarial transferability between models to defend against black-box adversarial attacks [79]). Concretely, ADP [53] needs the ground-truth labels to calculate the ensemble diversity. DVERGE [79] depends on the misalignment of the distilled feature between the visual similarity and the classification result, and hence on the imperceptible perturbation model. Therefore, it is non-trivial to directly adapt them to the discrete VRP domain. DivTrain [32] proposes to decrease the gradient similarity loss to reduce the overall adversarial transferability between models, and TRS [81] further uses a model smoothness loss to improve the ensemble robustness. Technically, their methods are computational expensive due to the needs to keep the computational graph before taking an optimization step. We show their empirical results in Table 7. Compared with others, their empirical results are not superior in VRPs. For example, for TSP100, POMO_TRS (3), which is adapted from [81], achieves 0.098% and 0.528% on the metrics of Uniform and Fixed Adv., respectively, failing to mitigate the undesirable trade-off. We leave the discussion on defensive methods from other domains (e.g., language and graph) to Section 2.

**Attack Budget.** As discussed in Section 3.2, we do not exert the imperceptible constraint on the perturbation model in VRPs. We further explain it from two perspectives: 1) different from other domains, there is no theoretical guarantee to ensure the invariance of the optimal solution (or ground-truth label), given an imperceptible perturbation. A small change on even one node's attribute may induce a totally different optimal solution. Therefore, we do not see the benefit of constraining the attack budget to a very small (i.e., imperceptible) range in VRP tasks. Moreover, even with the absence of the imperceptible constraint on the perturbation model, unlike the graph learning setting, we do not observe a significant degradation on clean performance. It reveals that we don't need explicit imperceptible perturbation to restrain the changes on (clean) objective values in VRPs. In our experiments, we set the attack budget within a reasonable range following the applied attack method (e.g., $\alpha \in [1, 100]$ for [87]). Our experimental results show that the proposed CNF is able to achieve a favorable balance, given different attack methods and their attack budgets; 2) in VRPs (or COPs), all generated adversarial instances are valid problem instances regardless of how much they differ from the clean instances [20]. In this sense, the attack budget models the severity of a potential distribution shift between training data and test data. This highlights the differences to other domains (e.g., vision), where unconstrained perturbations may lead to non-realistic or invalid data. Technically, the various attack budgets can help to generate diverse adversarial instances for training. Considering the above aspects, we believe our adversarial setting, including diverse but valid problem instances, may benefit the VRP community in developing a more general and robust neural solver.

With that said, this paper can also be viewed as an attempt to improve the generalization of neural VRP solvers from the perspective of adversarial robustness.

**Attack Scenarios and Practical Significance.** Actually, there may not be a person or attacker to deliberately invade the VRP model in practice. However, the developers in a logistics enterprise should consider the adversarial robustness of neural solvers as a sanity check before deploying them in the real world. The adversarial attack can be viewed as a way of measuring the worst-case performance of neural solvers within the neighborhood of inputs. Without considering adversarial robustness, the neural solver may perform very poorly (see Fig. 1(c)) when the testing instance pattern in the real world is 1) different from the training one, and 2) similar to the adversarial one. For example, considering an enterprise like Amazon, when some new customers need to be added to the current configuration or instance, especially when their locations coincidentally lead to the adversarial instance of the current solver (i.e., the node insertion attack presented in Appendix B.2), the model without considering adversarial robustness may output a very bad solution, and thus resulting in unpleasant user experience and financial losses. Furthermore, one may argue that this robustness issue can be ameliorated by using other solvers, i.e., NCO alongside traditional heuristics. We would like to note that traditional solvers may be non-robust as well, as discussed in a recent work [42]. The availability of traditional solvers cannot be guaranteed for novel problems as well.

**The Selection Basis of Attackers.** There are three attackers in the current VRP literature [87, 20, 42]. We select the attacker based on its generality. Specifically, 1) [20] is an early work that explicitly investigates the adversarial robustness in COPs. Their perturbation model needs to be sound and efficient, which means, given a clean instance and its optimal solution, the optimal solution to the adversarial instance can be directly derived without running a solver. However, this direct derivation requires unique properties and theorems of certain problems (e.g., the intersection theorem [9] for Euclidean-based TSP), and hence is non-trivial to generalize to more complicated VRPs (e.g., CVRP). Moreover, their perturbation model is limited to attack the supervised neural solver (i.e., ConvTSP [31]), since it needs to construct the adversarial instance by maximizing the loss function so that the model prediction is maximally different from the derived optimal solution. While in VRPs, RL-based methods [37, 38] are more appealing since they can gain comparable performance without the need for optimal solutions. 2) [42] requires that the optimal solution to the adversarial instance is no worse than that to the clean instance in theory, which may limit the search space of adversarial instances. It focuses on the graph-based COPs (e.g., asymmetric TSP) and satisfies the requirement by lowering the cost of edges. Similar to [20], their method is not easy to design for VRPs with more constraints. Moreover, they resort to the black-box adversarial attack method by training a reinforcement learning based attacker, which may lead to a higher computational complexity and relatively low success rate of attacking. Therefore, for better generality, we choose [87] as the attacker in the main paper, which can be applied to different VRP variants and popular VRP solvers [37, 38]. Moreover, we also evaluate the versatility of CNF against [42] as presented in Appendix D.3.

**Training Efficiency.** As mentioned in Section 6, the limitation of the proposed CNF is the increased computational complexity due to the need to synergistically train multiple models. Concretely, we empirically observe that CNF requires more training time than the vanilla AT variants on TSP100 (e.g., POMO_AT (3) 32 hrs vs. CNF (3) 85 hrs). However, simply further training them cannot significantly increase their performance since the training process has nearly converged. For example, given roughly the same training time, POMO_AT achieves 0.246% and 0.278% while CNF achieves 0.118% and 0.236% on the metrics of Uniform and Fixed Adv., respectively. On the other hand, our training time is less than that of advanced AT methods (e.g., POMO_TRS (3) [81] needs around 94 hrs). The above comparison indicates that CNF can achieve a better trade-off to deliver superior results within a reasonable computational budget. Moreover, we find that the global attack generation consumes much more training time than the neural router ($\sim 0.05$ million parameters). During inference, the computational complexity of CNF only depends on the number of models, since the neural router is only activated during training, and is discarded afterwards. Therefore, all methods with the same number of trained models have the same inference time.

**Relationship between OOD Generalization and Adversarial Robustness in VRPs.** Generally, the adversarial robustness measures the generalization capability of a model over the perturbed instance $\tilde{x}$ in the proximity of the clean instance $x$. In the context of VRPs (or COPs), adversarial instances are neither anomalous nor statistical defects since they are valid problem instances regardless of how much they differ from the clean instance $x$ [20]. In contrast to other domains (e.g., vision), 1) the attack budget models the severity of a potential distribution shift between training data and test

data; 2) the set of valid problems is not just a low-dimensional manifold in a high-dimensional space, and hence the manifold hypothesis [62] does not apply to combinatorial optimization. Therefore, it is critical for neural VRP solvers to perform well on adversarial instances when striving for a broader OOD generalization. Based on experimental results, we empirically demonstrate that raising robustness against attacks through CNF favorably promotes various forms of generalization of neural VRP solvers (as shown in Section 5.3), indicating the potential existence of neural VRP solvers with high generalization and robustness concurrently.

**Robustness of Test-time Adaptation Methods.** Test-time adaptation methods [26, 56, 82] are relatively robust. This degree of "robustness" primarily comes from 1) traditional local search heuristics and metaheuristics, which can effectively handle perturbations but bring much post-processing runtime, or 2) the costly fine-tuning that incurs more inference time on each test instance. Our CNF can also be combined with them to pursue better performance. Furthermore, we conduct experiments on DIMES [56] and DeepACO [82]. The results are shown in Tables 3 and 4, where we observe that 1) CNF outperforms them in most cases (e.g., w/o advanced search) with a much shorter runtime, and 2) test-time adaptation methods (i.e., the purple columns) improve robustness by adopting much post-search effort and consuming hours of runtime. According to the results, we note that 1) directly comparing CNF with these methods with advanced searches (e.g., MCTS, NLS) is somewhat unfair, since the additional post-processing search makes the analysis of model robustness difficult. For example, recent work finds that MCTS is so strong that using an almost all-zero heatmap can achieve good solutions [77], raising doubts about the real robustness of learned models; 2) the post-processing searches are typically time-consuming and problem-specific, making them non-trivial to be adapted to other COPs. Finally, we highlight that studies on robustness in COPs are still rare, and most work focused on investigating the robustness of neural solvers themselves, due to their advantages in generality on more complex COPs and inference speed. However, attacking test-time adaption methods is a potential research direction, which we will leave to future work.

Table 3: Robustness study of DIMES [56] on 1000 TSP100 instances. G, S, MCTS, and AS denote greedy, sample, Monte Carlo tree search, and active search, respectively.

| | DIMES (G) | | DIMES (AS+G) | | DIMES (S) | | DIMES (AS+S) | | DIMES (MCTS) | | DIMES (AS+MCTS) | |
| | Gap | Time | Gap | Time | Gap | Time | Gap | Time | Gap | Time | Gap | Time |
|---|---|---|---|---|---|---|---|---|---|---|---|---|
| Clean | 14.65% | 1.31m | 5.21% | 2.42h | 13.50% | 1.40m | 5.11% | 2.44h | 0.05% | 2.50m | 0.03% | 2.47h |
| Fixed Adv. | 19.29% | 1.32m | 12.12% | 2.45h | 18.24% | 1.42m | 11.87% | 2.45h | 0.19% | 2.50m | 0.16% | 2.49h |

Table 4: Robustness study of DeepACO [82] on 1000 TSP100 instances. Note that only DeepACO (NLS) uses local search. We use 100 ants and $T \in \{1, 10, 50, 100\}$ ACO iterations. The total inference time of each method with $T = 100$ is reported.

| | ACO - 1.7h | | | | DeepACO - 1.7h | | | | DeepACO (NLS) - 5.6h | | | |
| | T=1 | T=10 | T=50 | T=100 | T=1 | T=10 | T=50 | T=100 | T=1 | T=10 | T=50 | T=100 |
|---|---|---|---|---|---|---|---|---|---|---|---|---|
| Clean | 98.65% | 46.15% | 25.47% | 20.31% | 13.65% | 7.21% | 5.54% | 5.08% | 0.46% | 0.08% | 0.03% | 0.02% |
| Fixed Adv. | 40.87% | 19.09% | 12.61% | 10.92% | 25.50% | 14.60% | 10.75% | 9.71% | 0.31% | 0.08% | 0.00% | 0.00% |

# B    Attack Methods

In this section, we first give a formal definition of adversarial instance in VRPs, and then present details of existing attack methods for neural VRP solvers, including perturbing input attributes [87], inserting new nodes [20], and lowering the cost of a partial problem instance to ensure no-worse theoretical optimum [42]. They can generate instances that are underperformed by the current model.

We define an adversarial instance as the instance that 1) is obtained by the perturbation model within the neighborhood of the clean instance, and 2) is underperformed by the current model. Formally, given a clean VRP instance $x = \{x_c, x_d, \sigma\}$, where $x_c \in \mathcal{N}_c$ is the continuous variable (e.g., node coordinates) within the valid range $\mathcal{N}_c$; $x_d \in \mathcal{N}_d$ is the discrete variable (e.g., node demand) within the valid range $\mathcal{N}_d$; $\sigma$ is the constraint, the adversarial instance $\tilde{x} = \{\tilde{x}_c, \tilde{x}_d, \tilde{\sigma}\}$ is found by the perturbation model $G$ around the clean instance $x$, on which the current model may be vulnerable. Technically, the adversarial instance $\tilde{x}$ can be constructed by adding crafted perturbations $\gamma$ to the corresponding attribute of the clean instance, and then project them back to the valid domain, e.g., $\tilde{x}_c = \Pi_{\mathcal{N}_c}(x_c + \alpha \cdot \gamma_{x_c})$, where $\Pi$ denotes the projection operator (e.g., min-max normalization); $\alpha$ denotes the attack budget. The crafted perturbations can be obtained by various perturbation models, such as the one in Eq. (7), $\gamma_{x_c} = \nabla_{x_c} \ell(x; \theta)$, where $\theta$ is the model parameters; $\ell$ is the loss function. We omit the attack step $t$ for notation simplicity. An illustration of generated adversarial instances is shown in Fig. 4. Below, we follow the notations from the main paper, and detail each attack method.

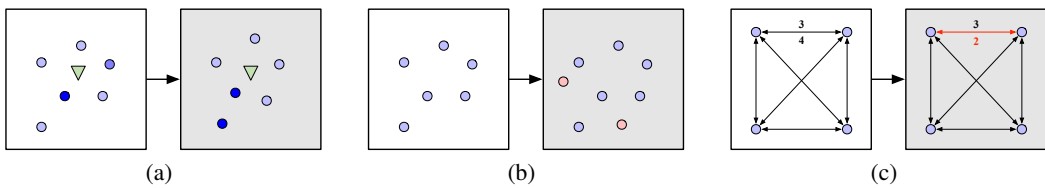

Figure 4: An illustration of generated adversarial instances (i.e., the grey ones). (a) An adversarial instance generated by [87] on CVRP, where the triangle represents the depot node. A deeper color denotes a heavier node demand; (b) An adversarial instance generated by [20] on TSP, where the red nodes represent the newly inserted adversarial nodes; (c) An adversarial instance generated by [42] on asymmetric TSP, where the cost of an edge is in half.

## B.1    Attribute Perturbation

The attack generator from [87] is applied to attention-based models [37, 38] by perturbing attributes of input instances. As introduced in Section 4, it generates adversarial instances by directly maximizing the reinforcement loss variant (so called the hardness measure in [87]). We take the perturbation on the node coordinate as an example. Suppose given the clean instance $x$ (i.e., $\tilde{x}^{(0)} = x$) and model parameter $\theta$, the solution to the inner maximization can be approximated as follows:

$$\tilde{x}^{(t+1)} = \Pi_{\mathcal{N}_c}[\tilde{x}^{(t)} + \alpha \cdot \nabla_{\tilde{x}^{(t)}} \ell(\tilde{x}^{(t)}; \theta^{(t)})], \tag{7}$$

where $\tilde{x}^{(t)}$ is the (local) adversarial instances and $\theta^{(t)}$ is the model parameters, at step $t$. Here we use $\mathcal{N}_c$ to represent the valid domain of node coordinates (i.e., unit square $U(0, 1)$) for simplicity. After each iteration, it checks whether $\hat{x}^{(t)} = \tilde{x}^{(t)} + \alpha \cdot \nabla_{\tilde{x}^{(t)}} \ell(\tilde{x}^{(t)}; \theta^{(t)})$ is within the valid domain $\mathcal{N}_c$ or not. If it is out of $\mathcal{N}_c$, the projection operator (i.e., min-max normalization) is applied as follows:

$$\Pi_{\mathcal{N}_c}(\hat{x}^{(t)}) = \frac{\hat{x}^{(t)} - \min \hat{x}^{(t)}}{\max \hat{x}^{(t)} - \min \hat{x}^{(t)}} (\max \mathcal{N}_c - \min \mathcal{N}_c). \tag{8}$$

Note that it originally only focuses on TSP, where the node coordinates are perturbed. We further adapt it to CVRP by perturbing both the node coordinates and node demands. The implementation is straightforward, except that we set the valid domain of node demands as $\mathcal{N}_d = \{1, \ldots, 9\}$. For the perturbations on node demands, the projection operator applies another round operation as follows:

$$\Pi_{\mathcal{N}_d}(\hat{x}^{(t)}) = \lceil \frac{\hat{x}^{(t)} - \min \hat{x}^{(t)}}{\max \hat{x}^{(t)} - \min \hat{x}^{(t)}} (\max \mathcal{N}_d - \min \mathcal{N}_d) \rceil. \tag{9}$$

## B.2 Node Insertion

An efficient and sound perturbation model is proposed by [20], which, given the optimal solution $y$ to the clean instance $x$ sampled from the data distribution $\mathcal{D}$, guarantees to directly derive the optimal solution $\tilde{y}$ to the adversarial instance $\tilde{x}$ without running a solver. The attack is applied to the GCN [31], which is a non-autoregressive construction method for TSP. It learns the probability of each edge occurring in the optimal solution (i.e., heat-map) with supervised learning. Following the AT framework, the objective function can be written as follows:

$$\min_{\theta} \mathbb{E}_{(x,y)\sim\mathcal{D}} \max_{\tilde{x}} \ell(f_{\theta}(\tilde{x}), \tilde{y}), \text{ with } \tilde{x} \in G(x,y) \wedge \tilde{y} = h(\tilde{x}, x, y), \tag{10}$$

where $\ell$ is the cross-entropy loss; $G$ is the perturbation model that describes the possible perturbed instances $\tilde{x}$ around the clean instance $x$; and $h$ is used to derive the optimal solution $\tilde{y}$ based on $(\tilde{x}, x, y)$ without running a solver. In the inner maximization, the adversarial instance $\tilde{x}$ is generated by inserting several new nodes into $x$ (below we take inserting one new node as an example), which adheres to below proposition and proof (by contradiction) borrowed from [20]:

**Proposition 1.** *Let $Z \notin \mathcal{V}$ be an additional node to be inserted, $w(\mathcal{E})$ is an edge weight, and $P, Q$ are any two neighbouring nodes in the original optimal solution $y$. Then, the new optimal solution $\tilde{y}$ (including $Z$) is obtained from $y$ through inserting $Z$ between $P$ and $Q$ if $\nexists(A,B) \in \mathcal{E} \setminus \{(P,Q)\}$ with $A \neq B$ s.t. $w(A,Z) + w(B,Z) - w(A,B) \leq w(P,Z) + w(Q,Z) - w(P,Q)$.*

**Proof.** Let $(R,S) \in \mathcal{E} \setminus \{(P,Q)\}$ to be two neighboring nodes of $Z$ on $\tilde{y}$. Assume $w(P,Z) + w(Q,Z) - w(P,Q) < w(R,Z) + w(S,Z) - w(R,S)$ and the edges $(P,Z)$ and $(Q,Z)$ are not contained in $\tilde{y}$ (i.e., $Z$ is inserted between $R$ and $S$ rather than $P$ and $Q$).

Below inequalities hold by the optimality of $y$ and $\tilde{y}$:

$$c(\tilde{y}) - w(R,Z) - w(S,Z) + w(R,S) \geq c(y). \tag{11}$$

$$c(y) + w(P,Z) + w(Q,Z) - w(P,Q) \geq c(\tilde{y}). \tag{12}$$

Therefore, we have

$$c(y) + w(P,Z) + w(Q,Z) - w(P,Q) \geq c(\tilde{y}) \geq c(y) + w(R,Z) + w(S,Z) - w(R,S), \tag{13}$$

which leads to a contradiction against the assumption (i.e., $w(P,Z) + w(Q,Z) - w(P,Q) < w(R,Z) + w(S,Z) - w(R,S)$). The proof is completed.

They use a stricter condition $\nexists(A,B) \in \mathcal{E} \setminus \{(P,Q)\}$ with $A \neq B$ s.t. $w(A,Z) + w(B,Z) - w(A,B) \leq w(P,Z) + w(Q,Z) - w(P,Q)$ in the proposition, since it is unknown which nodes can be $R$ and $S$ in $\tilde{y}$. Moreover, for the metric TSP, whose node coordinate system obeys the triangle inequality (e.g., euclidean distance), it is sufficient if the condition of Proposition 1 holds for $(A,B) \in \mathcal{E} \setminus (\{(P,Q)\} \cup \mathcal{H})$ with $A \neq B$ where $\mathcal{H}$ *denotes the pairs of nodes both on the Convex Hull $\mathcal{H} \in CH(\mathcal{E})$ that are not a line segment of the Convex Hull.* It is due to the fact that the optimal route $\tilde{y}$ must be a simple polygon (i.e., no crossings are allowed) in the metric space. This conclusion was first stated for the euclidean space as "the intersection theorem" by [9] and is a direct consequence of the triangle inequality.

Based on the above-mentioned proposition, the optimization of inner maximization involves: 1) obtaining the coordinates of additional node $Z$ by gradient ascending (e.g., maximizing $l$ such that the model prediction is maximally different from the derived optimal solution $\tilde{y}$); 2) penalizing if $Z$ violates the constraint in Proposition 1. Unfortunately, the constraint is non-convex and hard to find a relaxation. Instead of optimizing the Lagrangian (which requires extra computation for evaluating $f_{\theta}$), the vanilla gradient descent is leveraged with the constraint as the objective:

$$Z \leftarrow Z - \eta \nabla_Z [w(P,Z) + w(Q,Z) - w(P,Q) - (\min_{A,B} w(A,Z) + w(B,Z) - w(A,B))], \tag{14}$$

where $\eta$ is the step size. After we find $Z$ satisfying the constraint, the adversarial instance $\tilde{x}$ and the corresponding optimal solution $\tilde{y}$ can be constructed directly. Finally, the outer minimization takes $(\tilde{x}, \tilde{y})$ as inputs to train the robust neural solvers. This attack method can be easily adopted by our proposed framework, where $\theta$ is replaced by the best model $\theta_b$ when maximizing the cross-entropy loss (i.e., $\max_{\tilde{x}} \ell(f_{\theta_b}(\tilde{x}), \tilde{y})$) in the inner maximization optimization. However, due to the efficiency and soundness of the perturbation model, it does not suffer from the undesirable trade-off following the vanilla AT [20]. Therefore, we mainly focus on other attack methods [87, 42].

### B.3 No-Worse Theoretical Optimum

The attack method specialized for graph-based COPs is proposed by [42]. It resorts to the black-box adversarial attack method by training a reinforcement learning based attacker, and thus can be used to generate adversarial instances for both differentiable (e.g., learning-based) and non-differentiable (e.g., heuristic or exact) solvers. In this paper, we only consider the learning-based neural solvers for VRPs. Specifically, it generates the adversarial instance $\tilde{x}$ by modifying the clean instance $x$ (e.g., lowering the cost of a partial instance) under the no worse optimum condition, which requires $c(\tilde{y}) \leq c(y)$ if we are solving a minimization optimization problem. The attack is successful (w.r.t. the neural solver $\theta$) if the output solution to $\tilde{x}$ is worse than the one to $x$ (i.e., $c(\tilde{\tau}|\tilde{x};\theta) > c(\tau|x;\theta) \geq c(y) \geq c(\tilde{y})$). The training of the attacker is hence formulated as follows:

$$\max_{\tilde{x}}. \quad c(\tilde{\tau}|\tilde{x};\theta) - c(\tau|x;\theta),$$
$$\text{s.t.} \quad \tilde{x} = G(x, T; \theta), \ c(\tilde{y}) \leq c(y), \tag{15}$$

where $G(x, T; \theta)$ represents the deployment of the attacker $G$ trained on the given model (or solver) $\theta$ to conduct $T$ modifications on the clean instance $x$. It is trained with the objective as in Eq. (15) using the RL algorithm (i.e., Proximal Policy Optimization (PPO)). Specifically, the attack process is modelled as a MDP, where, at step $t$, the state is the current instance $\tilde{x}^{(t)}$; the action is to select an edge whose weight is going to be half; and the reward is the increase of the objective: $c(\tau^{(t+1)}|\tilde{x}^{(t+1)};\theta) - c(\tau^{(t)}|\tilde{x}^{(t)};\theta)$. This process is iterative until $T$ edges are modified. We use ROCO to represent this attack method in the remaining of this paper.

ROCO has been applied to attack MatNet [39] on asymmetric TSP (ATSP). As shown in the empirical results of [42], conducting the vanilla AT may suffer from the trade-off between standard generalization and adversarial robustness. To solve the problem, we further apply our method to defend against it. However, it is not straightforward to adapt ROCO to the inner maximization of CNF, since ROCO belongs to the black-box adversarial attack method, which does not directly rely on the parameters (or gradients) of the current model to generate adversarial instances. Concretely, we first train an attacker $G_j$ using RL for each model $\theta_j$, obtaining $M$ attackers for $M$ models ($\Theta = \{\theta_j\}_{j=0}^{M-1}$) in CNF. For the local attack, we simply use $G_j$ to generate local adversarial instances for $\theta_j$ by $G_j(x, T; \theta_j)$. For the global attack, we decompose the generation process (i.e., modifying $T$ edges) of a global adversarial instance $\bar{x}$ as follows:

$$\bar{x}^{(t+1)} = G_b^{(t)}(\bar{x}^{(t)}, 1; \theta_b^{(t)}), \quad \theta_b^{(t)} = \arg\min_{\theta \in \Theta} c(\tau|\bar{x}^{(t)};\theta), \tag{16}$$

where $\bar{x}^{(t)}$ is the global adversarial instance; $\theta_b^{(t)}$ is the best-performing model (w.r.t. $\bar{x}^{(t)}$); and $G_b^{(t)}$ is the attacker corresponding to $\theta_b^{(t)}$, at step $t \in [0, T-1]$. Since the model $\theta_j$ is updated throughout the optimization, to save the computation, we fix the attacker $G_j$ and only update (by retraining) it every $E$ epochs using the latest model. More details and results can be found in Appendix D.3.

## C Neural Router

### C.1 Model Structure

Without loss of generality, we take TSP as an example, where an instance consists of coordinates of $n$ nodes. The attention-based neural router takes as inputs $N$ instances $X \in \mathbb{R}^{N \times n \times 2}$ and a cost matrix $\mathcal{R} \in \mathbb{R}^{N \times M}$, where $M$ is the number of models in CNF. The neural router first embeds the raw inputs into h-dimensional (i.e., 128) features as follows:

$$F_I = \texttt{Mean}(W_1 X + b_1), \quad F_R = W_2 \mathcal{R} + b_2, \tag{17}$$

where $F_I \in \mathbb{R}^{N \times h}$ and $F_R \in \mathbb{R}^{N \times h}$ are features of instances and cost matrices, respectively; $W_1, W_2$ are weight matrices; $b_1, b_2$ are biases. The $\texttt{Mean}$ operator is taken along the second dimension of inputs. Then, a single-head attention layer (i.e., *glimpse* [1]) is applied:

$$Q = W_Q([F_I, F_R]), \quad K = W_K(\texttt{Emb}(M)), \tag{18}$$

where $[\cdot, \cdot]$ is the horizontal concatenation operator; $Q \in \mathbb{R}^{N \times h}$ is the query matrix; $K \in \mathbb{R}^{M \times h}$ is the key matrix; $W_Q, W_K$ are the weight matrices; $\texttt{Emb}(M) \in \mathbb{R}^{M \times h}$ is a learnable embedding layer

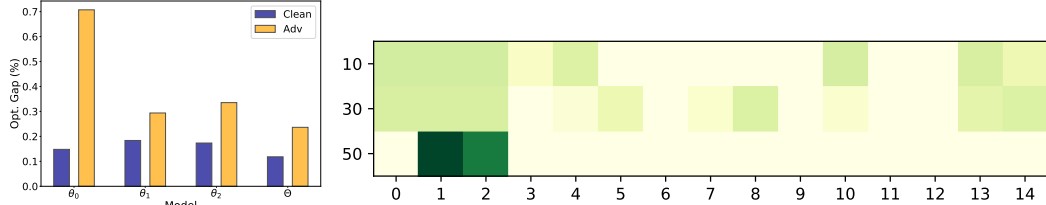

Figure 5: *Left panel:* Performance of each model $\theta_j \in \Theta$ in CNF ($M = 3$), and the overall collaboration performance of $\Theta$. *Right panel:* A demonstration (i.e., attention map) of the learned routing policy for $\theta_0$. The horizontal axis is the index of the training instance. Concretely, 0-2: clean instances $x$; 3-11: local adversarial instances $\tilde{x}$; 12-14: global adversarial instances $\bar{x}$. The vertical axis is the epoch of the checkpoint. A deeper color represents a higher probability to be selected.

representing the features of $M$ models. The logit matrix $\mathcal{P} \in \mathbb{R}^{N \times M}$ is calculated as follows:

$$\mathcal{P} = C \cdot \texttt{tanh}(\frac{QK^T}{\sqrt{h}}), \tag{19}$$

where the result is clipped by the `tanh` function with $C = 10$ following [1]. When the neural router is applied to CVRP, we only slightly modify $Q$ by further concatenating it with the features of the depot and node demands, while keeping others the same.

### C.2 Learned Routing Policy

We attempt to briefly interpret the learned routing policy. We first show the performance of each model $\theta_j \in \Theta$ in CNF in the left panel of Fig. 5, which is trained on TSP100 following the training setups presented in Section 5. Although not all models perform well on both clean and adversarial instances, the collaborative performance of $\Theta$ is quite good, demonstrating the diverse expertise of each model and the capability of CNF in reasonably exploiting the overall model capacity. We further give a demonstration (i.e., attention map) of the learned routing policy in the right panel of Fig. 5. We take the first model $\theta_0$ as an example, whose expertise lies in clean instances. For simplicity and readability of the results, the batch size is set to $B = 3$, and thus the number of input instances $\mathcal{X}$ for the neural router is 15. The neural router then distributes $B = 3$ instances to each model (if using model choice routing strategies). Note that the instances for different epochs are not the same, while the types remain the same (e.g., instances with ids 0-2 are clean instances). From the results, we observe that 1) the learned policy tends to distribute all types of instances to each model in a balanced way at the beginning of training, when the model is vulnerable to adversarial instances; 2) clean instances are more likely to be selected at the end of training, when the model is relatively robust to adversarial instances while trying to mitigate the accuracy-robustness trade-off.

## D Experiments

### D.1 Extra Setups

**Setups for Baselines.** We compare our method with several strong traditional and neural VRP methods. Following the conventional setups in the community [37, 38, 26], for specialized heuristic solvers such as Concorde, LKH3 and HGS, we run them on 32 CPU cores for solving TSP and CVRP instances in parallel, while running neural methods on one GPU card. Below, we provide the implementation details of baselines. 1) Concorde: We use Concorde Version 03.12.19 with the default setting, to solve TSP instances. 2) LKH3 [25]: We use LKH3 Version 3.0.8 to solve TSP and CVRP instances. For each instance, we run LKH3 with 10000 trails and 1 run. 3) HGS [70]: We run HGS with the default hyperparameters to solve CVRP instances. The maximum number of iterations without improvement is set to 20000. 4) For POMO [38], beyond the open-source pretrained model, we further train it using the vanilla AT framework (POMO_AT). Specifically, following the training setups presented in Section 5, we use the pretrained model to initialize $M$ models, and train them individually using local adversarial instances generated by themselves. 5) POMO_HAC [87] further

improves upon the vanilla AT. It constructs a mixed dataset with both clean instances and local adversarial instances for training afterwards. In the outer minimization, it optimizes an instance-reweighted loss function based on the instance hardness. Following their setups, the weight for each instance $x_i$ is defined as: $w_i = \exp(\mathcal{F}(\mathcal{H}(x_i))/\mathcal{T})/\sum_j \exp(\mathcal{F}(\mathcal{H}(x_j))/\mathcal{T})$, where $\mathcal{F}$ is the transformation function (i.e., tanh); $\mathcal{T}$ is the temperature controlling the weight distribution. It starts from 20 and decreases linearly as the epoch increasing. The hardness $\mathcal{H}$ is computed following Eq. (4). 6) POMO_DivTrain is adapted from the diversity training [32], which studies the ensemble-based adversarial robustness in the image domain by proposing a novel method to train an ensemble of models with uncorrelated loss functions. Specifically, it improves the ensemble diversity by minimizing the cosine similarity between the gradients of (sub-)models w.r.t. the input. Its loss function is formulated as: $\mathcal{L} = \ell + \lambda \log(\sum_{1 \leq a < b \leq M} \exp(\frac{<\nabla_x \ell_a, \nabla_x \ell_b>}{|\nabla_x \ell_a||\nabla_x \ell_b|}))$, where $\ell$ is the original loss function; $M$ is the number of models; $\nabla_x \ell_a$ is the gradient of the loss function (on the $a_{th}$ model) w.r.t. the input $x$; $\lambda = 0.5$ is a hyperparameter controlling the importance of gradient alignment during training. 7) CNF_Greedy: the neural router is simply replaced by the heuristic method, where each model selects $\mathcal{K}$ hardest instances. We use [87] as the attack method in Section 5. For training efficiency, we set $T = 1$ in the main experiments. The step size $\alpha$ is randomly sampled from 1 to 100.

**Setups for Ablation Study.** We conduct extensive ablation studies on components, hyperparameters and routing strategies as shown in Section 5. For simplicity, we slightly modified the training setups. We train all methods using 2.5M TSP100 instances. The learning rate is decayed by 10 for the last 20% training instances. For the ablation on components (Fig. 3(a)), we set the attack steps as $T = 2$, and remove each component separately to demonstrate the effectiveness of each component in our proposed framework. For the ablation on hyperparameters (Fig. 3(b)), we train multiple models with $M \in \{2, 3, 4, 5\}$. For the ablation on routing strategies (Fig. 3(c)), we set $\mathcal{K} = B$ for M-Top$\mathcal{K}$, where $B = 64$ is the batch size, and $\mathcal{K} = 1$ for I-Top$\mathcal{K}$. The other training setups remain the same.

**Setups for Versatility Study.** For the pretraining stage, we train MatNet [39] on 5M ATSP20 instances following the original setup from [39]. We further train a perturbation model by attacking it using reinforcement learning. Concretely, we use the dataset from [42], consisting of 50 "tmat" class ATSP training instances that obey the triangle inequality, to train the perturbation model for 500 epochs. Adam optimizer is used with the learning rate of $1e - 3$. The maximum number of actions taken by the perturbation model is $T = 10$. After the pretraining stage, we use the pretrained model to initialize $M = 3$ models, and further adversarially train them. We fix the perturbation model and only update it using the latest model every $E = 10$ epochs (as discussed in Appendix B.3). After the $10_{\text{th}}$ epoch, there would be $M$ perturbation models corresponding to $M$ models. Following [42], we use the fixed 1K clean instances for training. In the inner maximization, we generate $M$K local adversarial instances and 1K global adversarial instances using the perturbation models. However, since the perturbation model is not efficient (i.e., it needs to conduct beam search to find edges to be modified), we generate adversarial instances in advance and reuse them later. Then, in the outer minimization, we load all instances and distribute them to each model using the jointly trained neural router. The models are then adversarially trained for 20 epochs using the Adam optimizer with the learning rate of $4e - 5$, the weight decay of $1e - 6$, and the batch size of $B = 100$.

### D.2  Data Generation

We follow the instructions from [37] to generate synthetic instances. Concretely, 1) **Uniform Distribution:** The node coordinate of each node is uniformly sampled from the unit square $U(0, 1)$, as shown in Fig. 6(a). 2) **Rotation Distribution:** Following [6], we mutate the nodes, which originally follow the uniform distribution, by rotating a subset of them (anchored in the origin of the Euclidean plane) as shown in Fig. 6(b). The coordinates of selected nodes are transformed by multiplying them with a matrix $\begin{bmatrix} \cos(\varphi) & \sin(\varphi) \\ -\sin(\varphi) & \cos(\varphi) \end{bmatrix}$, where $\varphi \sim [0, 2\pi]$ is the rotation angle. 3) **Explosion Distribution:** Following [6], we mutate the nodes, which originally follow the uniform distribution, by simulating a random explosion. Specifically, we first randomly select the center of explosion $v_c$ (i.e., the hole as shown in Fig. 6(c)). All nodes $v_i$ within the explosion radius $R = 0.3$ is moved away from the center of explosion with the form of $v_i = v_c + (R + s) \cdot \frac{v_c - v_i}{||v_c - v_i||}$, where $s \sim \text{Exp}(\lambda = 1/10)$ is a random value drawn from an exponential distribution.

In this paper, we mainly consider the distribution of node coordinates. For CVRP instances, the coordinate of the depot node $v_0$ is uniformly sampled from the unit square $U(0, 1)$. The demand

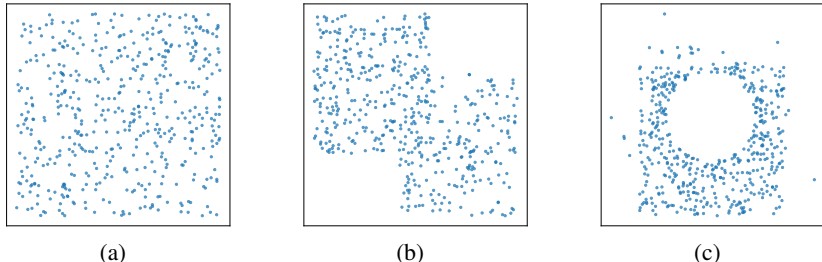

Figure 6: The generated TSP instances following the (a) Uniform distribution; (b) Rotation distribution; (c) Explosion distribution.

of each node $\delta_i$ is randomly sampled from a discrete uniform distribution $\{1, \ldots, 9\}$. The capacity of each vehicle is set to $Q = \lceil 30 + \frac{n}{5} \rceil$, where $n$ is the size of a CVRP instance. The demand and capacity are further normalized to $\delta_i' = \delta/Q$ and 1, respectively.

Table 5: Performance evaluation against ROCO [42] over 1K ATSP instances.

| | Clean | | | | Fixed Adv. | | | |
|---|---|---|---|---|---|---|---|---|
| | (x1) Gap | Time | (x16) Gap | Time | (x1) Gap | Time | (x16) Gap | Time |
| LKH3 | 0.000% | 1s | 0.000% | 1s | 0.000% | 1s | 0.000% | 1s |
| Nearest Neighbour | 30.481% | – | 30.481% | – | 31.595% | – | 31.595% | – |
| Farthest Insertion | 3.373% | – | 3.373% | – | 3.967% | – | 3.967% | – |
| MatNet (1) | 0.784% | 0.5s | 0.056% | 5s | 0.931% | 0.5s | 0.053% | 5s |
| MatNet_AT (1) | 0.817% | 0.5s | 0.072% | 5s | 0.827% | 0.5s | 0.046% | 5s |
| MatNet_AT (3) | 0.299% | 1.5s | 0.028% | 15s | 0.319% | 1.5s | 0.023% | 15s |
| CNF (3) | **0.246%** | 1.5s | **0.022%** | 15s | **0.278%** | 1.5s | **0.015%** | 15s |

## D.3 Versatility Study

We demonstrate the versatility of CNF by applying it to MatNet [39] to defend against another attack [42]. The results are shown in Table 5, where we evaluate all methods on 1K ATSP instances. For neural methods, we use the sampling with x1 and x16 instance augmentations following [39, 42]. The gaps are computed w.r.t. LKH3. Based on the results, we observe that 1) CNF is effective in mitigating the undesirable accuracy-robustness trade-off; 2) together with the main results in Section 5, CNF is versatile to defend against various attacks among different neural VRP methods.

Table 6: Sensitivity Analyses on hyperparameters.

| Remark | Optimizer | Batch Size | Normalization | LR | Uniform (100) | Fixed Adv. (100) |
|---|---|---|---|---|---|---|
| Default | Adam | 64 | Instance | 1e-4 | 0.111% | 0.255% |
| | SGD | 64 | Instance | 1e-4 | 0.146% | 3.316% |
| | Adam | 32 | Instance | 1e-4 | 0.122% | 0.262% |
| | Adam | 128 | Instance | 1e-4 | **0.088%** | 0.311% |
| | Adam | 64 | Batch | 1e-4 | 0.114% | **0.247%** |
| | Adam | 64 | Instance | 1e-3 | 0.183% | 0.282% |
| | Adam | 64 | Instance | 1e-5 | 0.101% | 0.616% |

## D.4 Sensitivity Analyses

In addition to the ablation study on the key hyperparameter (i.e., the number of models $M$), we further conduct sensitivity analyses on others, such as the optimizer $\in$ [Adam, SGD], batch size $\in$ [32, 64, 128], normalization layer $\in$ [batch, instance], and learning rate (LR) $\in$ [$1e{-}3, 1e{-}4, 1e{-}5$]). The experiments are conducted on TSP100 following the setups of the ablation study as presented in Appendix D.1. The results are shown in Table 6, where we observe that the performance of our method can be further boosted by carefully tuning hyperparameters.

## D.5 Ablation Study on CVRP

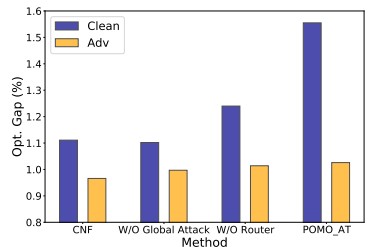

Figure 7: Ablation study on Components.

We further conduct the ablation study on CVRP. For simplicity, here we only consider investigating the role of each component in CNF by removing them separately. The experiments are conducted on CVRP100, and the setups are kept the same as the ones presented in Appendix D.1. The results are shown in Fig. 7, which still verifies the effectiveness of the global attack and neural router in CNF.

## D.6 Advanced Ensemble-based AT

Here, we consider several advanced ensemble-based AT methods, such as ADP [53], DVERGE [79], TRS [81]. However, as we discussed in Appendix A, it is non-trivial to adapt ADP and DVERGE to the VRP domain due to their needs for ground-truth labels or the dependence on the imperceptible perturbation model. Therefore, we compare our method with TRS [81], which improves upon Div-Train [32]. Concretely, it proposes to use the gradient similarity loss and another model smoothness loss to improve ensemble robustness. We follow the experimental setups presented in Section 5, and show the results of TSP100 in Table 7. We observe that TRS achieves better standard generalization but much worse adversarial robustness, and hence fail to achieve a satisfactory trade-off in our setting. Moreover, TRS is more computationally expensive than CNF in terms of memory consumption, since it needs to keep the computational graph for all submodels before taking an optimization step.

Table 7: Comparison with advanced ensemble-based AT on TSP100.

|  | POMO (1) | POMO_AT(3) | POMO_HAC (3) | POMO_DivTrain (3) | POMO_TRS (3) | CNF (3) |
|---|---|---|---|---|---|---|
| Uniform (100) | 0.144% | 0.255% | 0.135% | 0.255% | **0.098%** | 0.118% |
| Fixed Adv. (100) | 35.803% | 0.295% | 0.344% | 0.297% | 0.528% | **0.236%** |

## D.7 Visualization of Adversarial Instances

The distribution of generated adversarial instances depends on the attack method and its strength. Here, we take the attacker from [87] as an example. We visualize clean instances and their corresponding adversarial instances in Fig. 8, where we show distribution shifts of node coordinates and node demands on CVRP100, respectively.

# E Broader Impacts

Recent works have highlighted the vulnerability of neural VRP methods to adversarial perturbations, with most research focusing on the generation of adversarial instances (i.e., attacking). In contrast, this paper delves into the adversarial defense of neural VRP methods, filling the gap in the current literature on this topic. Our goal is to enhance model robustness and mitigate the undesirable accuracy-robustness trade-off, which is a commonly observed phenomenon in adversarial ML. We propose an ensemble-based collaborative neural framework to adversarially train multiple models in a collaborative manner, achieving high generalization and robustness concurrently. Our work sheds light on the possibility of building more robust and generalizable neural VRP methods in practice. However, since our approach introduces extra operations upon the vanilla AT, such as the global attack in the inner maximization and the neural router in the outer minimization, the computation cost may not be friendly to our environment. Therefore, exploring more efficient and scalable attack or defense methods for VRPs (or COPs) is a worthwhile endeavor for future research.

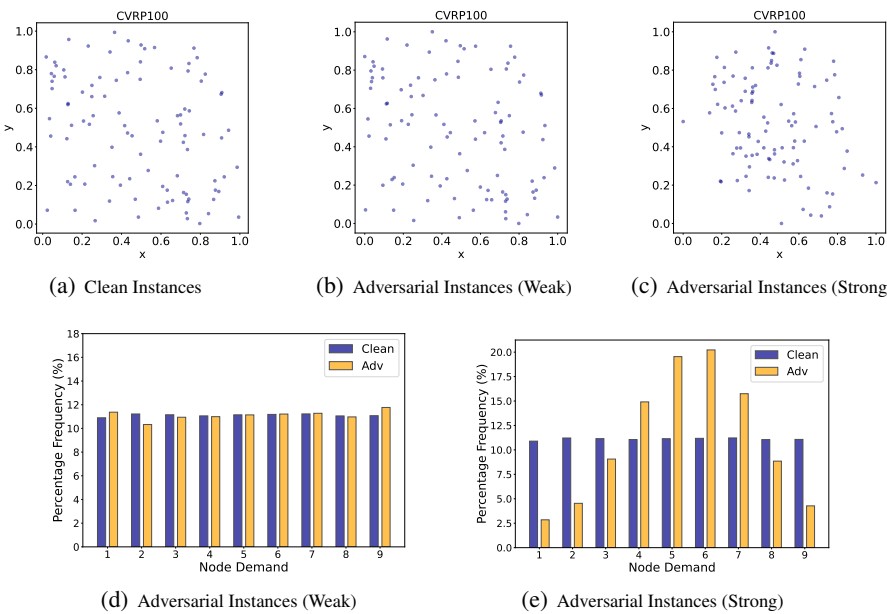

(a) Clean Instances     (b) Adversarial Instances (Weak)     (c) Adversarial Instances (Strong)

(d) Adversarial Instances (Weak)     (e) Adversarial Instances (Strong)

Figure 8: Visualization of clean instances and corresponding adversarial instances generated by weak and strong attack [87]. (a-c) shows the spatial distribution of node locations, and (d-e) shows the percentage frequency distribution of node demands over the entire CVRP100 test dataset.

## F    Licenses

Table 8: List of licenses for code and datasets used in this work.

| Resource | Type | Link | License |
|---|---|---|---|
| Concorde | Code | https://www.math.uwaterloo.ca/tsp/concorde.html | Available for academic research use |
| LKH3 [25] | Code | http://webhotel4.ruc.dk/~keld/research/LKH-3 | Available for academic research use |
| HGS [70] | Code | https://github.com/vidalt/HGS-CVRP | MIT License |
| POMO [38] | Code | https://github.com/yd-kwon/POMO | MIT License |
| MatNet [39] | Code | https://github.com/yd-kwon/MatNet | MIT License |
| DIMES [56] | Code | https://github.com/DIMESTeam/DIMES | MIT License |
| DeepACO [82] | Code | https://github.com/henry-yeh/DeepACO | MIT License |
| HAC [87] | Code | https://github.com/wondergo2017/tsp-hac | No License |
| ROCO [42] | Code | https://github.com/Thinklab-SJTU/ROCO | No License |
| TRS [81] | Code | https://github.com/AI-secure/Transferability-Reduced-Smooth-Ensemble | No License |
| TSPLIB [58] | Dataset | http://comopt.ifi.uni-heidelberg.de/software/TSPLIB95 | Available for any non-commercial use |
| CVRPLIB [67] | Dataset | http://vrp.galgos.inf.puc-rio.br/index.php | Available for academic research use |

Table 9: Results on TSPLIB [58] instances. Models are only trained on $n = 100$.

| Instance | Opt. | POMO | | POMO_AT | | POMO_HAC | | POMO_DivTrain | | CNF | |
|---|---|---|---|---|---|---|---|---|---|---|---|
| | | Obj. | Gap | Obj. | Gap | Obj. | Gap | Obj. | Gap | Obj. | Gap |
| kroA100 | 21282 | 21420 | 0.65% | 21347 | 0.31% | **21308** | **0.12%** | 21370 | 0.41% | **21308** | **0.12%** |
| kroB100 | 22141 | 22200 | 0.27% | 22211 | 0.32% | 22200 | 0.27% | **22199** | **0.26%** | 22216 | 0.34% |
| kroC100 | 20749 | 20799 | 0.24% | 20768 | 0.09% | **20753** | **0.02%** | 20768 | 0.09% | 20758 | 0.04% |
| kroD100 | 21294 | 21446 | 0.71% | 21391 | 0.46% | 21407 | 0.53% | 21435 | 0.66% | **21353** | **0.28%** |
| kroE100 | 22068 | 22259 | 0.87% | 22288 | 1.00% | 22167 | 0.45% | 22213 | 0.66% | **22121** | **0.24%** |
| eil101 | 629 | 630 | 0.16% | 630 | 0.16% | **629** | **0.00%** | 631 | 0.32% | 630 | 0.16% |
| lin105 | 14379 | 14477 | 0.68% | 14426 | 0.33% | 14408 | 0.20% | **14402** | **0.16%** | 14403 | 0.17% |
| pr107 | 44303 | 44678 | 0.85% | 47819 | 7.94% | **44596** | **0.66%** | 46285 | 4.47% | 44719 | 0.94% |
| pr124 | 59030 | 59389 | 0.61% | 59257 | 0.38% | 59385 | 0.60% | 59558 | 0.89% | **59076** | **0.08%** |
| bier127 | 118282 | 133042 | 12.48% | 118606 | 0.27% | 118608 | 0.28% | **118337** | **0.05%** | 118841 | 0.47% |
| ch130 | 6110 | 6119 | 0.15% | 6130 | 0.33% | 6115 | 0.08% | 6125 | 0.25% | **6111** | **0.02%** |
| pr136 | 96772 | 97983 | 1.25% | 100225 | 3.57% | 97617 | 0.87% | 100145 | 3.49% | **97567** | **0.82%** |
| pr144 | 58537 | 58935 | 0.68% | 59544 | 1.72% | 58913 | 0.64% | 59265 | 1.24% | **58868** | **0.57%** |
| ch150 | 6528 | 6554 | 0.40% | 6582 | 0.83% | 6556 | 0.43% | 6578 | 0.77% | **6550** | **0.34%** |
| kroA150 | 26524 | 26755 | 0.87% | 26898 | 1.41% | 26736 | 0.80% | 26813 | 1.09% | **26722** | **0.75%** |
| kroB150 | 26130 | 26405 | 1.05% | 26506 | 1.44% | **26379** | **0.95%** | 26467 | 1.29% | 26494 | 1.39% |
| pr152 | 73682 | **74249** | **0.77%** | 77537 | 5.23% | 75291 | 2.18% | 77127 | 4.68% | 74876 | 1.62% |
| rat195 | 2323 | 2486 | 7.02% | 2500 | 7.62% | 2461 | 5.94% | 2467 | 6.20% | **2449** | **5.42%** |
| kroA200 | 29368 | 29992 | 2.12% | 30222 | 2.91% | 29771 | 1.37% | 30143 | 2.64% | **29755** | **1.32%** |
| kroB200 | 29437 | 30298 | 2.92% | 30157 | 2.45% | 29890 | 1.54% | 30267 | 2.82% | **29862** | **1.44%** |
| ts225 | 126643 | 134609 | 6.29% | 135801 | 7.23% | **128085** | **1.14%** | 134569 | 6.26% | 128436 | 1.42% |
| tsp225 | 3916 | 4035 | 3.04% | 4031 | 2.94% | 4004 | 2.25% | 4025 | 2.78% | **4001** | **2.17%** |
| pr226 | 80369 | **83470** | **3.86%** | 89455 | 11.31% | 85466 | 6.34% | 90347 | 12.42% | 84914 | 5.66% |
| pr264 | 49135 | 55157 | 12.26% | 60390 | 22.91% | 53894 | 9.69% | 60957 | 24.06% | **53763** | **9.42%** |
| a280 | 2579 | 2740 | 6.24% | 2741 | 6.28% | **2690** | **4.30%** | 2760 | 7.02% | 2701 | 4.73% |
| pr299 | 48191 | 50785 | 5.38% | 50812 | 5.44% | 50487 | 4.76% | 50535 | 4.86% | **50408** | **4.60%** |
| lin318 | 42029 | 43430 | 3.33% | 43808 | 4.23% | **42860** | **1.98%** | 43840 | 4.31% | 43060 | 2.45% |
| rd400 | 15281 | 15775 | 3.23% | 16221 | 6.15% | 15755 | 3.10% | 16135 | 5.59% | 15778 | 3.25% |
| fl417 | 11861 | **13958** | **17.68%** | 15240 | 28.49% | 14544 | 22.62% | 15637 | 31.84% | 14317 | 20.71% |
| pr439 | 107217 | 123357 | 15.05% | 120545 | 12.43% | 117963 | 10.02% | 120212 | 12.12% | **117632** | **9.71%** |
| pcb442 | 50778 | 54087 | 6.52% | 54686 | 7.70% | **53165** | **4.70%** | 55125 | 8.56% | 53281 | 4.93% |
| d493 | 35002 | 64215 | 83.46% | 38356 | 9.58% | 37685 | 7.67% | 38168 | 9.05% | 38051 | 8.71% |
| u574 | 36905 | 41456 | 12.33% | 42045 | 13.93% | 40804 | 10.57% | 41990 | 13.78% | **40737** | **10.38%** |
| rat575 | 6773 | 7828 | 15.58% | 7774 | 14.78% | 7658 | 13.07% | 7791 | 15.03% | **7635** | **12.73%** |
| p654 | 34643 | 46094 | 33.05% | 51915 | 49.86% | **45447** | **31.19%** | 52837 | 52.52% | 46425 | 34.01% |
| d657 | 48912 | 59082 | 20.79% | 56232 | 14.97% | 55580 | 13.63% | 56635 | 15.79% | **55066** | **12.58%** |
| u724 | 41910 | 49171 | 17.33% | 50583 | 20.69% | 48818 | 16.48% | 50679 | 20.92% | **48692** | **16.18%** |
| rat783 | 8806 | 10819 | 22.86% | 10769 | 22.29% | 10512 | 19.37% | 10767 | 22.27% | **10473** | **18.93%** |
| pr1002 | 259045 | 325734 | 25.74% | 328459 | 26.80% | 322276 | 24.41% | 335954 | 29.69% | **320624** | **23.77%** |

Table 10: Results on CVRPLIB [67] instances. Models are only trained on $n = 100$.

| Instance | Opt. | POMO | | POMO_AT | | POMO_HAC | | POMO_DivTrain | | CNF | |
|---|---|---|---|---|---|---|---|---|---|---|---|
| | | Obj. | Gap | Obj. | Gap | Obj. | Gap | Obj. | Gap | Obj. | Gap |
| X-n101-k25 | 27591 | 29282 | 6.13% | 28919 | 4.81% | 29176 | 5.74% | 29019 | 5.18% | **28911** | **4.78%** |
| X-n106-k14 | 26362 | 26961 | 2.27% | **26608** | **0.93%** | 26789 | 1.62% | 26679 | 1.20% | 26672 | 1.18% |
| X-n110-k13 | 14971 | 15154 | 1.22% | 15298 | 2.18% | 15305 | 2.23% | **15125** | **1.03%** | 15127 | 1.04% |
| X-n120-k6 | 13332 | 14574 | 9.32% | 13762 | 3.23% | 13734 | 3.02% | 13801 | 3.52% | **13652** | **2.40%** |
| X-n134-k13 | 10916 | 11315 | 3.66% | **11189** | **2.50%** | 11250 | 3.06% | 11259 | 3.14% | 11248 | 3.04% |
| X-n143-k7 | 15700 | 16382 | 4.34% | 16233 | 3.39% | 16019 | 2.03% | 16192 | 3.13% | **15980** | **1.78%** |
| X-n148-k46 | 43448 | 47613 | 9.59% | 46546 | 7.13% | 46433 | 6.87% | 46504 | 7.03% | **45694** | **5.17%** |
| X-n162-k11 | 14138 | 14986 | 6.00% | 14923 | 5.55% | 14827 | 4.87% | 14942 | 5.69% | **14794** | **4.64%** |
| X-n181-k23 | 25569 | 26969 | 5.48% | 26282 | 2.79% | 26299 | 2.86% | **26211** | **2.51%** | 26213 | 2.52% |
| X-n195-k51 | 44225 | 50296 | 13.73% | 50228 | 13.57% | 48987 | 10.77% | 51367 | 16.15% | **48823** | **10.40%** |
| X-n214-k11 | 10856 | 11752 | 8.25% | 11868 | 9.32% | **11570** | **6.58%** | 11760 | 8.33% | 11587 | 6.73% |
| X-n233-k16 | 19230 | 21107 | 9.76% | 21054 | 9.49% | 20997 | 9.19% | 21067 | 9.55% | **20980** | **9.10%** |
| X-n251-k28 | 38684 | 41355 | 6.90% | 41313 | 6.80% | 41193 | 6.49% | 41212 | 6.54% | **40992** | **5.97%** |
| X-n270-k35 | 35291 | 39952 | 13.21% | 38837 | 10.05% | **38376** | **8.74%** | 38511 | 9.12% | 38411 | 8.84% |
| X-n289-k60 | 95151 | 105343 | 10.71% | 104923 | 10.27% | **104236** | **9.55%** | 104786 | 10.13% | 104261 | 9.57% |
| X-n294-k50 | 47161 | 53937 | 14.37% | 53772 | 14.02% | 52876 | 12.12% | 53789 | 14.05% | **52699** | **11.74%** |
| X-n313-k71 | 94043 | 105470 | 12.15% | 105647 | 12.34% | 104179 | 10.78% | 104375 | 10.99% | **103726** | **10.30%** |
| X-n331-k15 | 31102 | 42292 | 35.98% | 39293 | 26.34% | 36957 | 18.83% | 37047 | 19.11% | **36194** | **16.37%** |
| X-n376-k94 | 147713 | 162224 | 9.82% | **157156** | **6.39%** | 158410 | 7.24% | 158168 | 7.08% | 157880 | 6.88% |
| X-n384-k52 | 65928 | 76139 | 15.49% | 76527 | 16.08% | 76080 | 15.40% | 78141 | 18.52% | **74878** | **13.58%** |
| X-n439-k37 | 36391 | 46077 | 26.62% | 48372 | 32.92% | 49018 | 34.70% | 46614 | 28.09% | **45219** | **24.26%** |
| X-n469-k138 | 221824 | **252278** | **13.73%** | 264366 | 19.18% | 258635 | 16.59% | 263698 | 18.88% | 258751 | 16.65% |
| X-n502-k39 | 69226 | 85692 | 23.79% | 83326 | 20.37% | 83702 | 20.91% | 80082 | 15.68% | **79373** | **14.66%** |

