# OpenReview forum: "Collaboration! Towards Robust Neural Methods for Routing Problems"
_NeurIPS.cc/2024/Conference — NeurIPS 2024 poster_

### Official Review · Reviewer_d2Cb · 2024-06-18

**Soundness:** 3
**Presentation:** 3
**Contribution:** 3
**Rating:** 6
**Confidence:** 3

**Summary:**

This paper focuses on the robustness of neural combinatorial optimization and proposes a new framework to robustify neural models for routing problems. The core idea is to train a set of models and to let them collectively make the best predictions. The authors have conducted experiments to benchmark the proposed method against various baselines for TSP and CVRP.

**Strengths:**

S1: The approach to emsembling multiple models in this paper seems novel and interesting. I believe that the idea will be interesting even for more general combinatorial optimization problems.

S2: The empirical performance of the method is impressive. The proposed method consistently outperforms all baselines on TSP and CVRP. Besides that, the experiments section is well written. Sections E.2 (data generation) and 4.3 (discussion on OOD generalization) are particularly interesting.

S3: The authors have provided code and detailed description of experimental settings (Section E.1) to facilitate reproducibility.

**Weaknesses:**

W1: The proposed method is collaborative AT. Hence, a missing ablation study is *independent* AT, i.e., using AT to train $M$ models independently (say, under different seeds) and outputting the best solutions found by the $M$ models.

W2: This paper did not describe the threat model considered in this work. There should be no guarantee of performance if arbitrary perturbations are allowed. Thus, it might be better if the authors can clearly state what types of perturbations the proposed method is robust against.

W3: This paper describes mainly the training procedure but barely the inference procedure. It might be better if the paper can elaborate more on the detailed inference procedure.

Minor issue: While the authors (as well as prior work) use "attack" throughout the paper, I personally do not think this is a precise term. The setting in this paper seems more like OOD than attacking because I am unable to come up with a real-world scenario where an attacker wants to perturb the instance only in order to make the model produce a worse solution for the *perturbed* instance.

**Questions:**

Q1: How robust are test-time adaptation methods such as meta-learning [58] and neural-guided local search [89]? Since such methods do not solely rely on the pretrained model, they may be inherently robust. Hence, it might be better if this paper also compare with such test-time adaptation methods.
- [58] Qiu et al. DIMES: A differentiable meta solver for combinatorial optimization problems. NeurIPS, 2022.
- [89] Ye et al. DeepACO: Neural-enhanced ant systems for combinatorial optimization. NeurIPS, 2023.

Q2 (regarding the limitation): Since it is observed that the proposed method takes more time to converge, is it possible to employ better training schemes to speed up the convergence of the proposed method? For example, [58] found that meta-learning can significantly speed up convergence (see their Figure 1).

**Limitations:**

The authors have clearly discussed the main limitation in Section 5. Although using multiple models potentially increases the computational burden in both training and inference phases, the authors found that three models typically suffice to achieve good performance. Thus, this does not seem to be a severe limitation. See also Q2.

---

> ### Author Rebuttal · Authors · 2024-08-06
>
> We genuinely appreciate the effort the reviewer dedicated to evaluating our paper and providing insightful feedback! Here are our detailed responses to your comments, where `W` denotes Weakness (W1-W4) and `Q` denotes Question (Q1-Q2).
>
> **W1: An experiment on independent AT.**
>
> Actually, the baseline method POMO_AT (3) in Table 1 refers to the independent AT, where 3 models are adversarially trained separately, and output the best solution. The results of training more (yet independent) AT models can be found in Fig. 1, showing that our method achieves a better trade-off between computation and performance.
>
> **W2: This paper did not describe the considered threat model.**
>
> Specifically, we consider the threat model of [1] in the main paper, which means that our trained model is robust against the perturbation of input attributes (e.g., node coordinates). To demonstrate the versatility of our method, we consider another threat model of [2] in Appendix E.3, which can be viewed as the perturbation of constraints. Due to the page limit, we present the details of each threat model in Appendix C (e.g., C.1 for [1]; C.3 for [2]). Please kindly refer to them.
>
> **W3: Elaboration on the inference procedure.**
>
> Generally, given a single trained model, we construct the solution using the greedy rollout with x8 augmentations following POMO [3]. Given multiple ($M)$ models, we construct one solution per model following the above procedure, obtain $M$ solutions for each test instance, and report the best one. The clean performance is evaluated on the test dataset whose distribution is the same as the training one. The adversarial performance is evaluated on the adversarial dataset, which is generated by attacking the pretrained or current model. If we use the threat model of [1], adversarial instances are generated by Eqs. (7-8) as shown in Appendix C.1, and $θ^{(t)}$ in Eq. (7) is then adjusted for different methods and evaluation metrics (e.g., $θ^{(t)}$ is set to the parameters of the pretrained model $θ_p$ if the metric of `Fixed Adv.` is evaluated). Please see lines 256-269 for inference setups. We will include more inference details in the final draft.
>
> **W4: Minor issue: The "attack" term, and the real-world scenario.**
>
> We understand the reviewer's concern. This term is from the general ML and thus we keep it for consistency in this paper. Generally, the attack procedure can be viewed as a way of generating instances on which the current model underperforms. We discuss the potential attack scenarios and practical significance in Appendix A (lines 686-700). We will clarify this terminology and highlight its difference in the COP domain in the introduction section. Please let us know if you have any better recommendations.
>
> **Q1: How robust are test-time adaptation methods such as meta-learning and neural-guided local search?**
>
> Thanks for the interesting question. Yes, these test-time adaptation methods [4-6] are relatively robust. This degree of "robustness" primarily comes from 1) traditional local search heuristics and metaheuristics, which can effectively handle perturbations but bring much post-processing runtime, or 2) the costly fine-tuning that incurs more inference time on each test instance. Our CNF can also be combined with them to pursue better performance. For example, we further combine CNF with EAS [6] and deliver better TSPLIB results as shown in Table 1 of the [uploaded PDF](https://openreview.net/attachment?id=drriUYPVZV&name=pdf).
>
> Furthermore, following the reviewer's suggestion, we conduct experiments on DIMES [4] and DeepACO [5]. The results are shown in Tables 2 and 3 of the [uploaded PDF](https://openreview.net/attachment?id=drriUYPVZV&name=pdf), where we observe that 1) CNF (clean: 0.12%; adv: 0.24%) outperforms them in most cases (e.g., w/o advanced search) with a much shorter runtime, and 2) test-time adaptation methods (i.e., the purple columns) improve robustness by adopting much post-search effort and consuming hours of runtime. According to the results, we note that 1) directly comparing CNF with these methods with advanced searches (e.g., MCTS, NLS) is somewhat unfair, since the additional post-processing search makes the analysis of model robustness difficult. For example, recent work finds that MCTS is so strong that using an almost all-zero heatmap can achieve good solutions [7], raising doubts about the real robustness of learned models; 2) the post-processing searches are typically time-consuming and problem-specific, making them non-trivial to be adapted to other COPs.
>
> Finally, we highlight that studies on robustness in COPs are still rare, and most work focused on investigating the robustness of neural solvers themselves, due to their advantages in generality on more complex COPs and inference speed. However, inspired by the reviewer's question, attacking test-time adaption methods is a potential research direction, which we will leave to future work.
>
> **Q2: Is it possible to employ meta-learning to speed up the convergence?**
>
> RL-based AM/POMO is known for its slow convergence due to the autoregressive decoding and sparse reward problem. DIMES [4] designs a compact continuous parameterization space of the underlying distribution of candidate solutions (i.e., heatmap) to efficiently search for high-quality solutions. Therefore, DIMES shows a faster convergence (due to its novel paradigm and lightweight network) in their Fig. 1. Furthermore, meta-learning is typically leveraged to help the model generalize across tasks (e.g., problem instances in DIMES). The second-order meta-learning (e.g., MAML [8]) is computationally expensive for training POMO. Therefore, we experiment with FOMAML [8], a first-order meta-learning method. We set the inner update step as 1, and follow our original training setups. Unfortunately, we do not observe an improved training efficiency or convergence. However, we will further explore this inspiring idea in the future.

---

> > ### Comment · Reviewer_d2Cb · 2024-08-07
> >
> > Thanks for your detailed response, which adequately addresses my concerns. I have raised my score.

---

> > > ### Author Response · Authors · 2024-08-07
> > > **Thanks for your prompt feedback**
> > >
> > > We would like to thank the reviewer for the prompt feedback and raising the score!

---

> ### Author Response · Authors · 2024-08-06
> **References**
>
> ```
> [1] Learning to solve travelling salesman problem with hardness-adaptive curriculum. In AAAI 2022.
> [2] ROCO: A general framework for evaluating robustness of combinatorial optimization solvers on graphs. In ICLR 2023.
> [3] POMO: Policy Optimization with Multiple Optima for Reinforcement Learning. In NeurIPS 2020.
> [4] DIMES: A differentiable meta solver for combinatorial optimization problems. In NeurIPS 2022.
> [5] DeepACO: Neural-enhanced ant systems for combinatorial optimization. In NeurIPS 2023.
> [6] Efficient active search for combinatorial optimization problems. In ICLR 2022.
> [7] Position: Rethinking Post-Hoc Search-Based Neural Approaches for Solving Large-Scale Traveling Salesman Problems. In ICML 2024.
> [8] Model-agnostic meta-learning for fast adaptation of deep networks. In ICML 2017.
> ```

---

### Official Review · Reviewer_xLJf · 2024-06-22

**Soundness:** 3
**Presentation:** 2
**Contribution:** 2
**Rating:** 5
**Confidence:** 4

**Summary:**

This paper presents an ensemble-based Collaborative Neural Framework for for vehicle routing problems to improve robustness. Multiple models are adversarially trained for better robustness against attacks and improve generalization, and a neural router routes instances to models for better load balancing and efficacy.

**Strengths:**

Not clear.

**Weaknesses:**

1. The paper is not organized or written  well.
2. This paper is not technically sound.

**Questions:**

1. Figure 1 is not clearly explained and not referred in the paper.
2. During inference, "The best solution among them is returned", how can you tell which is the best solution?
3. What is "collaborative manner" exactly?
4. What distribution will the generated adversarial instances follow?
5. Does this paper focus on solving combinatorial optimization problems? How?
6. As an important part of evaluation, there is no discussion on Table 7 or 8.

**Limitations:**

Adequate.

---

> ### Author Rebuttal · Authors · 2024-08-06
>
> We are grateful to the reviewer for dedicating your precious time to review our paper and offer feedback. Here are our detailed responses to your comments, where `W` denotes Weakness and `Q` denotes Question (Q1-Q6).
>
> **W: The paper is not organized or written well; This paper is not technically sound.**
>
> Due to the space limit, please see the [General Response](https://openreview.net/forum?id=YfQA78gEFA&noteId=drriUYPVZV). We will further polish our paper.
>
> **Q1: Figure 1 is not clearly explained and not referred.**
>
> Fig. 1 is referred in line 34 of Section 1. The motivation of Fig. 1 is to show the robustness issue of existing neural combinatorial solvers and the effectiveness of our CNF compared to vanilla adversarial training (AT). The empirical observations of Fig. 1 can be summarized as follows:
>
> * Based on Fig. 1 (b-c): Recent neural combinatorial solvers (e.g., POMO) suffer from severe robustness issues - their performance significantly deteriorates on adversarial instances (e.g., >> 10% optimality gap on TSP100).
> * Based on Fig. 1 (a-b): Vanilla AT (i.e., POMO_AT (1)) can significantly improve the adversarial robustness of the base model (i.e., (b): ~35.80% -> ~0.39%), but at the sacrifice of standard generalization (i.e., (a): 0.14% -> 0.37%). This is the so-called trade-off between standard generalization and adversarial robustness.
> * Based on Fig. 1 (a-b): Simply increasing the model capacity by naive ensembling multiple models through AT cannot efficiently mitigate this trade-off (e.g., POMO_AT (9) ensembles 9 models, but the standard generalization is still worse than POMO (1)). In contrast, our CNF can effectively mitigate this trade-off by only using 3 models.
>
> **Q2: During inference, how can you tell which is the best solution?**
>
> Different from Bayesian optimization, evaluating the solution quality is simple and efficient in routing problems. After constructing multiple solutions, the solution with the minimal objective value (i.e., tour length) is the best one.
>
> **Q3: What is collaborative manner exactly?**
>
> Note that previous baselines (e.g., POMO_AT ($M$)) are **independent**, which means $M$ models are independently trained, and output the best solution found by them. With that said, there is no interaction among $M$ models during training. In contrast, our CNF leverages the **collaboration** mechanism, which can be explained from the below perspectives:
>
> * *Adversarial instance generation in the inner maximization:* we propose the global attack for synergizing multiple models to generate the global adversarial instance by attacking the best-performing model in each iteration, rather than only leveraging each model to independently generate their own local adversarial instances. Suppose we have $M=3$ models ($\Theta=\{\theta_1, \theta_2, \theta_3\}$), the traditional local adversarial instance $\tilde{x}^{(T)}$ for $\theta_1$ is generated through $T$ iterations, and $\theta_1$ is repeatedly used in each iteration. In contrast, our global adversarial instance $\bar{x}^{(T)}$ can be generated by: $\bar{x}^{(0)} \xrightarrow{\theta_3} \bar{x}^{(1)} \xrightarrow{\theta_1} ... \xrightarrow{\theta_2} \bar{x}^{(T)}$ (i.e., $\theta_3$ performs the best for $\bar{x}^{(0)}$; $\theta_1$ performs the best for $\bar{x}^{(1)}$; ...). Therefore, there is a collaboration among models for instance generation in CNF.
> * *Policy optimization in the outer minimization:* for baselines, each model is solely trained on the local adversarial instance generated by itself. We design a neural router, intending to maximize the improvement of ensemble performance, to adaptively forward training instances to different models for effective training. In this way, we empirically observe that each model tends to have its own expert area (as shown in Fig. 5), and the total model capacity can be reasonably exploited, which is also a kind of collaboration.
>
> In summary, CNF facilitates the interaction (or collaboration) among models, facilitating diverse instance generation and making better use of the limited model capacity, hence significantly outperforming baselines.
>
> **Q4: What distribution will the generated adversarial instances follow?**
>
> It depends on the attack method and its strength. We have provided an illustration in Fig. 4 of the paper. We further visualize the generated adversarial instances in Fig. 1 of the [uploaded PDF](https://openreview.net/attachment?id=drriUYPVZV&name=pdf), where we show distribution shifts of node coordinates and node demands on CVRP100, respectively.
>
> **Q5: Does this paper focus on solving COPs? How?**
>
> Not exactly. This paper focuses on *improving the robustness and generalization of recent neural combinatorial solvers through AT*. Most of these neural solvers leverage the attention mechanism and the variant of Transformer to solve VRPs (see Section 2.1 regarding how they solve VRPs). Nevertheless, their performance greatly degrades on adversarial instances. Therefore, this paper proposes *a new training framework* to adversarially train these neural solvers in a collaborative way, aiming to enhance their robustness and generalization for solving VRPs.
>
> **Q6: There is no discussion on Table 7 or 8.**
>
> We summarize the results in lines 326-329. Concretely, the results of Tables 7 and 8 show that our method outperforms baselines on most benchmark instances, which has two implications: 1) raising robustness against attacks through CNF can favorably promote various forms of generalization (e.g., cross-size and cross-distribution); 2) there may exist a neural solver with high generalization and robustness concurrently in the COP domain. With that said, the generalization and robustness may not be conflicting goals in COPs [1], which is different from other domains such as image-based tasks. We will add the above discussion in the final draft.
>
> ```
> [1] Generalization of neural combinatorial solvers through the lens of adversarial robustness. In ICLR 2022.
> ```

---

> > ### Author Response · Authors · 2024-08-12
> >
> > Dear Reviewer xLJf:
> >
> > The deadline for the author-reviewer discussion is approaching. Please kindly let us know if our response resolves your concerns and we appreciate it if you could give us any feedback. We thank you again for your precious time and effort.

---

> > > ### Comment · Reviewer_xLJf · 2024-08-13
> > >
> > > Thanks for the authors' response, which clarified some points. I raised my ranking accordingly.

---

> > > > ### Author Response · Authors · 2024-08-13
> > > > **Thanks for your feedback**
> > > >
> > > > We sincerely appreciate the reviewer for acknowledging our response and raising the score!

---

> > ### Author Response · Authors · 2024-08-13
> >
> > Dear Reviewer xLJf,
> >
> > Thank you once again for your insightful comments and helpful suggestions. As the author-reviewer discussion will end soon (< 24 hours from now), we would greatly appreciate it if you could take a moment to review our rebuttal. Please let us know if you have any further questions or concerns. We sincerely appreciate your time and effort.
> >
> > Best regards,
> > Authors

---

### Official Review · Reviewer_AuRq · 2024-06-29

**Soundness:** 3
**Presentation:** 3
**Contribution:** 3
**Rating:** 6
**Confidence:** 5

**Summary:**

This paper develops a ensemble-based method CNF for adversarial training to enhance the robustness of NCO solvers. CNF demonstrates outstanding performance compared to all current adversarial training frameworks.

**Strengths:**

1. The effectiveness of CNF is outstanding.
2. The motivation, statement, and objective notation in the article are relatively complete.
3. The paper is well-written.

**Weaknesses:**

1. The trade-off between generalization (on clean instances) and transitional robustness demonstrated in Figure is not clear.
2. Compared to the textual illustration of the proposed CNF, Figures 2 and Algorithm 1 are not intuitive enough.

**Questions:**

1. For CVRP, Is it possible to perform adversarial training on the demand values as well?
2. What's the setting of ablation models in the part Ablation on Components?
3. For TSP, is CNF equally effective in different network architectures (such as ELG [1])?
4. Can the neural router used to assign the testing datas to facilitate inference?
5. Minor questions:
Line 82 should be 100% instead of 100.
The notation $\tilde{x}$ and $\bar{x}$ is quite similar which may cause misunderstanding.
Section 2.2 should include more references, e.g., [2].
\theta{r} should be included as input in Algorithm 1.
What's the formula of b(x) in Eq. (4)?
﻿
[1] Gao, Chengrui, et al. ""Towards generalizable neural solvers for vehicle routing problems via ensemble with transferrable local policy."" arXiv preprint arXiv:2308.14104 (2023).
[2] Goodfellow, Ian, et al. ""Generative adversarial nets."" Advances in neural information processing systems 27 (2014).

**Limitations:**

The discussion in article is comprehensive enough.

---

> ### Author Rebuttal · Authors · 2024-08-06
>
> We greatly appreciate the reviewer's time and effort in reviewing our paper and providing constructive comments. Here are our detailed responses to your review, where `W` denotes Weakness and `Q` denotes Question (Q1-Q5).
>
> **W: The trade-off demonstrated in Figure is not clear; Figures 2 and Algorithm 1 are not intuitive enough.**
>
> Thanks for your comments. We will polish them to be clearer. Based on the results from Fig. 1 (a-b), conducting vanilla adversarial training (i.e., POMO_AT (1)) can significantly improve the adversarial robustness of the base model (POMO (1)), i.e., ~35% -> ~0.4%, but at the sacrifice of standard generalization, i.e., ~0.15% -> ~0.36%. This demonstrates the existence of an undesirable trade-off between standard generalization and adversarial robustness in VRPs. We further give a detailed explanation regarding Fig. 1, which will be included in the final draft. Please see our response to `Reviewer xLJf Q1`.
>
> **Q1: For CVRP, Is it possible to perform adversarial training on the demand values as well?**
>
> Yes, a round operation can be used to project perturbed demands into the discrete valid domain. Please see the detailed formulation (i.e., Eq. (7)(9)) in Appendix C.1. We further show the percentage frequency distribution shift of node demands if perturbing node demands of CVRP instances. Please see Fig. 1 (d-e) of the [uploaded PDF](https://openreview.net/attachment?id=drriUYPVZV&name=pdf) for details.
>
> **Q2: What's the setting of ablation models in the part Ablation on Components?**
>
> The detailed training setups of ablation studies can be found in lines 969-976 of Appendix E.1. We consider the case of $M=3$ models, and remove each component separately to demonstrate the effectiveness of each component in our proposed CNF. Concretely, our CNF is equivalent to the base model, augmented by the global attack and neural router (i.e., `CNF = POMO_AT + Global Attack + Router`). 1) If we remove the global attack (i.e., `W/O Global Attack`), then only the local adversarial instances are generated in the inner maximization. They are then distributed by the neural router in the outer minimization. 2) If we remove the neural router (i.e., `W/O Router`), then the global and local adversarial instances are generated in the inner maximization. In the outer minimization, the local adversarial instances are distributed to the corresponding model that generates them, while the global adversarial instances are randomly distributed to models for training. 3) If we remove both components, then it is the vanilla adversarial training (i.e., `POMO_AT`), where there is no interaction or collaboration among models.
>
> **Q3: For TSP, is CNF equally effective in different network architectures?**
>
> Yes, the proposed CNF can be naturally adapted to other construction solvers, such as ELG [1] and LEHD[2]. In this paper, we mainly consider POMO and MatNet to demonstrate the versatility of CNF to defend against various attacks on different VRPs. Following the reviewer's suggestion, we further conduct an experiment of ELG with a simplified training setup (i.e., half the training instances) due to the tight rebuttal period. The pretrained ELG achieves 0.23% and 15.24% on the clean instances and adversarial instances of TSP100, respectively. Collaboratively training three ELG models through CNF significantly improves the standard generalization and adversarial robustness concurrently (i.e., clean: 0.11%; adv.: 0.25%). This demonstrates the generality of CNF in improving the robustness of neural VRP solvers.
>
> **Q4: Can the neural router be used to assign the testing data to facilitate inference?**
>
> Yes, the neural router can be used during inference. The positive side is that it can save inference time if the number of models M is extremely large. However, the neural router cannot fully guarantee to select the corresponding best-performing model w.r.t. each test instance. It may also suffer from the generalization issue if facing OOD instances. With that said, its performance is generally inferior to our default inference strategy. So this is a trade-off between computation and performance. Based on the experiments, we find that $M=3$ models are sufficient for CNF, so we decided to deactivate the neural router to maximize performance during inference.
>
> **Q5: About minor questions.**
>
> We thank the reviewer for the careful review and for bringing these issues to our attention! We will fix them and add the references in the final draft. The formulation of the POMO baseline in Eq. (4) is $b(x)=\frac{1}{N}\sum_{i=1}^Nc(\tau_i)$, where $N$ is the number of start nodes, $c(\cdot)$ is the cost function, and $\tau$ is the constructed solution (or tour) for the input instance $x$.
>
> ```
> [1] Towards generalizable neural solvers for vehicle routing problems via ensemble with transferrable local policy. In IJCAI 2024.
> [2] Neural Combinatorial Optimization with Heavy Decoder: Toward Large Scale Generalization. In NeurIPS 2023.
> ```

---

> > ### Comment · Reviewer_AuRq · 2024-08-08
> >
> > Thank you for your response. I raised my score to 6.

---

> > > ### Author Response · Authors · 2024-08-08
> > > **Thanks for your prompt feedback**
> > >
> > > We appreciate the reviewer for the prompt feedback and acknowledgment of our rebuttal!

---

### Official Review · Reviewer_DziC · 2024-07-14

**Soundness:** 3
**Presentation:** 3
**Contribution:** 2
**Rating:** 6
**Confidence:** 2

**Summary:**

The paper proposes a Collaborative Neural Framework (CNF) to improve the robustness of neural methods for vehicle routing problems (VRPs) against adversarial attacks. CNF introduces two key innovations: first, a global attack strategy that generates diverse adversarial instances by attacking the best-performing model, in addition to the local attacks on individual models. Second, an attention-based neural router that effectively distributes instances to different models for training, achieving satisfactory load balancing and collaborative performance. Extensive experiments show the effectiveness and generalization ability of the proposed method against different attacks.

**Strengths:**

- The paper tackles the crucial yet underexplored problem of adversarial defense for neural VRP methods.
- The paper is well-structured, with clear problem formulation, technical details, and comprehensive experimental evaluations.

**Weaknesses:**

- The paper is primarily focused on the empirical evaluation of the proposed framework, without providing any theoretical analysis or guarantees on the robustness or generalization properties of the approach

**Questions:**

Please see the weakness part.

**Limitations:**

Yes, the authors discuss the limitations in the conclusion section.

---

> ### Author Rebuttal · Authors · 2024-08-06
>
> We sincerely appreciate the reviewer for spending valuable time reviewing our paper and providing positive comments. Here are our detailed responses to your comments, where `W` denotes Weakness.
>
> **W: Theoretical analysis or guarantees.**
>
> Thanks for your comment. The majority of the work within the "learning for solving VRP" literature focused on the empirical side. They prioritized closing the gap between neural solvers and traditional solvers, lacking theoretical proof of the improvement. In this situation, it’s challenging for us to provide a solid theoretical analysis within the tight rebuttal period. We will try to add further analysis in the final draft. Here, we would like to give some intuition and empirical proof for the effectiveness and soundness of the proposed method:
>
> * *The global attack makes CNF diverse to benefit policy exploration, and it is stronger than the local attack in terms of attacking the ensemble of models.* 1) Diverseness: The global attack is a generalization of the local attack, which means the local attack could be viewed as a kind of global attack when the same model is repeatedly used to generate adversarial examples in each attack iteration. Therefore, by leveraging different model parameters during the attack, the adversarial instances generated by global adversaries can be more diverse than those only generated by local adversaries. 2) Strongness: The collaborative performance of CNF depends on the best-performing model $θ_b$ (w.r.t. each instance) since its solution will be chosen as the final solution during inference. In CNF, the goal of inner maximization is to construct adversarial instances that can successfully fool the entire framework. Intuitively, if we choose to attack other models rather than $θ_b$, the constructed adversarial instances may not successfully fool $θ_b$, and therefore the final solution to the adversarial instance could still be good, which contradicts the goal of the inner maximization. Therefore, to increase the success rate of attacking the CNF framework, for each clean instance, we propose the global adversary and choose the corresponding best-performing model $θ_b$ to be attacked in each iteration of the inner maximization. To empirically justify it, we record the training dynamics of CNF (3) at the middle stage of training on TSP100. We observe that the global adversary could maximally degrade the model performance by increasing the optimality gap to 0.52% while the local adversary could only degrade the model to 0.37%. Please note that a smaller optimality gap denotes a weaker attack.
>
> * *The neural router empowers CNF to reasonably exploit the overall model capacity.* Intuitively, by distributing instances to suitable models for training, each model might be stimulated to have its own expert area. Accordingly, the overlap of their vulnerability areas may be decreased, which could promote the collaborative performance of CNF. As shown in Fig. 5, not all models perform well on each kind of instance. Such diversity in model expertise contributes to the mitigation of the trade-off between standard generalization and adversarial robustness, thereby significantly outperforming vanilla AT with multiple models.
>
> In summary, instead of independently training multiple models, the effectiveness of the proposed collaboration mechanism in CNF can be attributed to its diverse adversarial data generation and the reasonable exploit of overall model capacity. As shown in the ablation study (Fig. 3(a)), the diverse adversarial data generation is helpful in further improving the adversarial robustness (see results of `CNF vs. W/O Global Attack`). Meanwhile, the neural router has a bigger effect in mitigating the trade-off (see results of `CNF vs. W/O Router`). Both components contribute to the power of CNF, thereby significantly outperforming other baselines.

---

> > ### Author Response · Authors · 2024-08-13
> >
> > Dear Reviewer DziC,
> >
> > Thank you once again for your insightful comments and helpful suggestions. As the author-reviewer discussion will end soon (< 24 hours from now), we would greatly appreciate it if you could take a moment to review our rebuttal. Please let us know if you have any further questions or concerns. We sincerely appreciate your time and effort.
> >
> > Best regards,
> > Authors

---

### Author Rebuttal · Authors · 2024-08-06

## **General Response**

We extend our heartfelt gratitude to all reviewers for their valuable comments. We are pleased to see that the reviewers have recognized our research topic is **crucial** (`DziC`), our method is **novel** (`d2Cb`) and **effective** (`AuRq, d2Cb`), our paper is **well-written** (`DziC, AuRq, d2Cb`), and our experiments are **detailed** and **comprehensive** (`DziC, d2Cb`). We have carefully considered all reviewers' comments by providing detailed **[clarifications and answers]** to the raised concerns and questions, and conducting **[extra experiments]**:

* adapting CNF to ELG to demonstrate the generality;
* showing the distribution of generated adversarial instances;
* checking the robustness of test-time adaptation methods, and comparing CNF with them;
* training CNF with meta-learning to investigate whether it could speed up the convergence.

We will include the above content in the final draft. Furthermore, for the convenience of reviewers and AC to understand and evaluate our paper, **especially in response to the weaknesses raised by `Reviewer xLJf`**, we provide a comprehensive overview of our paper below.

----

**[Paper Organization]:** Section 1 introduces the motivation of the research topic and methodology, and summarizes our contributions. Section 2 elaborates on the background of VRPs and AT. Section 3 details our method, including the overview, technical details, and its soundness. Section 4 presents experimental settings, empirical results, insights, and analyses. The conclusion, limitations, and future work are finally stated in Section 5. The related work is reviewed in the Appendix due to the page limit.

**[Motivation of Research Topic]:** The study on robustness in combinatorial optimization problems (COPs) is critical. It is highly related to the practicality and reliability of neural combinatorial solvers. Recent works observe that the performance of existing neural combinatorial solvers would significantly degrade on adversarial instances (i.e., clean instances with crafted perturbations), and thus start paying more attention to robustness. Most of them only investigate the attack methods, leaving the exploration of effective defense underexplored. In this paper, we focus on not only effectively defending against various attacks but also largely improving the performance on clean instances. Such a study may benefit the development of versatile neural solvers in real-world scenarios.

**[Motivation of Methodology]:** Previous works simply use vanilla adversarial training (AT) from the adversarial ML literature to deploy defense, which causes an undesirable trade-off between standard generalization (on clean instances) and adversarial robustness (on adversarial instances). This issue is also a well-known research problem in adversarial ML, and an effective solution is to increase the model capacity (e.g., ensembling multiple models). However, according to our experiments in Fig. 1, independently training multiple models and then ensembling them cannot efficiently achieve a favorable trade-off. Therefore, it further motivates us to develop an ensemble-based *Collaborative Neural Framework (CNF)*, a more elegant AT framework facilitating collaboration among models, to achieve a better trade-off.

 **[Technical Soundness / Why CNF Works]:** Instead of independently training multiple models, the effectiveness of CNF can be attributed to the collaboration mechanism inherent in its *diverse adversarial data generation (i.e., global attack)* and *reasonable exploitation of overall model capacity (i.e., neural router)*. As shown in the ablation study (Fig. 3(a)), the diverse adversarial data generation is helpful in further improving the adversarial robustness (see results of `CNF vs. W/O Global Attack`). Meanwhile, the neural router has a bigger effect in mitigating the trade-off (see results of `CNF vs. W/O Router`). We refer to our response to `Reviewer DziC` for the intuition and empirical proof of the effectiveness and soundness of these two components.

Briefly, the global attack makes CNF diverse and strong in benefiting policy exploration and attacking the ensemble of models. The neural router empowers CNF to reasonably exploit the overall model capacity, stimulating each model to have its own expert area. Such diversity in data generation and model expertise contributes to the mitigation of the trade-off between standard generalization and adversarial robustness, thereby significantly outperforming previous baselines.

**[Insight and Significance]:** *Conceptually*, we present insights into the differences in AT between the continuous image domain and the discrete COP domain in Section 2.2. *Technically*, we present insights into how AT could be leveraged to improve the robustness and generalization of neural combinatorial solvers concurrently. *Empirically*, we present insights about 1) the effectiveness of AT as a way of improving the robustness and generalization of neural combinatorial solvers. Raising robustness against attacks through CNF can favorably promote various forms of generalization (e.g., cross-size and cross-distribution); 2) the existence of neural combinatorial solvers with both high generalization and robustness. This implies that generalization and robustness may not be conflict goals in the COP domain.

In summary, our paper presents an early attempt to study the adversarial robustness of neural combinatorial solvers in the context of VRPs. It is also the first work to effectively improve the standard generalization and adversarial robustness concurrently, taking a step towards a more robust and generalizable neural combinatorial solver. We believe our insights can significantly benefit the robustness study in the NCO community, and serve as a solid avenue for future work.

----

We hope our responses can address all reviewers' concerns, and we welcome further discussions if any point is unclear.

The Authors

---

### Decision · Program_Chairs · 2024-09-25

**Decision:**

Accept (poster)

**Comment:**

After the rebuttal, reviewers think that their concerns have been mostly mitigated and they all agree that the paper has benefits. In particular, reviewers think that the paper tackles an important open problem of adversarial defense for routing problems and the effectiveness is outstanding. Therefore, AC will follow the reviewers' vote to accept the paper.

However, note that reviewers also mention some remaining issues that should be addressed in the revised version to further improve the paper. These include:

1. The paper lacks theoretical analysis and guarantee: please provide theoretical evidence for the following two arguments as authors mentioned in the rebuttal: "The global attack makes CNF diverse to benefit policy exploration, and it is stronger than the local attack in terms of attacking the ensemble of models" "The neural router empowers CNF to reasonably exploit the overall model capacity."

2. Please polish your description about the trade-off between generalization and transitional robustness as well as the intuition behind Figures 2 and Algorithm 1.

3. Please clarify the threat models and inference details in the main body.

Please take the above-mentioned comments into consideration in the next version of the paper.